# Loss of microglial SIRPα promotes synaptic pruning in preclinical models of neurodegeneration

Xin Ding [1,4], Jin Wang[1,2,4], Miaoxin Huang[1,4], Zhangpeng Chen[1,4], Jing Liu[1], Qipeng Zhang [1,3], Chenyu Zhang[1], Yang Xiang[1], Ke Zen [1✉] & Liang Li [1,3✉]

Microglia play a key role in regulating synaptic remodeling in the central nervous system. Activation of classical complement pathway promotes microglia-mediated synaptic pruning during development and disease. CD47 protects synapses from excessive pruning during development, implicating microglial SIRPα, a CD47 receptor, in synaptic remodeling. However, the role of microglial SIRPα in synaptic pruning in disease remains unclear. Here, using conditional knock-out mice, we show that microglia-specific deletion of SIRPα results in decreased synaptic density. In human tissue, we observe that microglial SIRPα expression declines alongside the progression of Alzheimer's disease. To investigate the role of SIRPα in neurodegeneration, we modulate the expression of microglial SIRPα in mouse models of Alzheimer's disease. Loss of microglial SIRPα results in increased synaptic loss mediated by microglia engulfment and enhanced cognitive impairment. Together, these results suggest that microglial SIRPα regulates synaptic pruning in neurodegeneration.

[1] Nanjing Drum Tower Hospital Center of Molecular Diagnostic and Therapy, Chinese Academy of Medical Sciences Research Unit of Extracellular RNA, State Key Laboratory of Pharmaceutical Biotechnology, Nanjing University, Nanjing, China. [2] Department of Endocrinology, Drum Tower Hospital Affiliated to Nanjing University Medical School, Nanjing, China. [3] Institute for Brain Sciences, Nanjing University, Nanjing, China. [4] These authors contributed equally: Xin Ding, Jin Wang, Miaoxin Huang, Zhangpeng Chen. ✉email: kzen@nju.edu.cn; liangli@nju.edu.cn

Substantial evidence has demonstrated that microglia play a pivotal role in synapse remodeling[1–3]. Dysfunction of microglia-mediated synaptic pruning during neurodevelopment results in abnormal social behavior in adulthood[4,5]. Furthermore, such malfunction is also implicated in several disease-associated neurological symptoms[6–8]. Removing the redundant synapses correctly is important for the normal development and functional homeostasis of the healthy brain, which requires collaboration of both positive and negative signals modulating synaptic elimination[9].

Previous studies mainly focused on the signals that positively regulate microglia-mediated synaptic pruning[10,11]. The innate immune molecules, C1q and C3, have been demonstrated to promote microglial engulfment of synaptic structures via activating C3 receptor (CR3)[12,13]. Another microglial chemokine receptor, Cx3cr1, is also required for microglial pruning during neurodevelopment and whose deficiency undermines neural refinement and cause social behavior defects[2,14]. These signals not only control synapse remodeling in early neurodevelopment, but also play crucial roles in several neurological diseases[13,15,16]. Alzheimer's Disease (AD) is an aging related neurodegenerative disorder, in which synapse loss is the major correlate of cognitive impairment[17]. It has been demonstrated that C1q level is significantly enhanced in neuronal synapses in the early stage of AD, resulting in the consequent synaptic loss, while blocking the signal significantly alleviates neurodegeneration in AD mice model[18]. The abnormality of complement system may also cause neurological symptoms in several other diseases such as virus infection and schizophrenia, accompanied by aberrant microglia-mediated synapse loss[16,19,20]. These results suggest that appropriate microglia-mediated synapse elimination is important for maintaining normal brain function.

Despite much research conducted on the signals that positively modulate synaptic clearance, the negative regulatory pathways involved in microglia-mediated synaptic pruning are less known. Until recently, Lehrman et al.[9] have discovered that neuronal CD47, which can bind to SIRPα and produce "do not eat me" signal, protects synapses from excessive elimination. CD47 null mice have less synapses with increased microglial pruning in dorsal lateral geniculate nucleus (dLGN), indicating the potential protective role of SIRPα-CD47 axis in synapses refinement during neurodevelopment[9].

SIRPα (also known as SHPS-1, p84, and BIT) is a transmembrane protein that binds to its ligand, CD47, activating the protein tyrosine phosphatases SHP-1 or SHP-2 through its cytoplasmic region[21]. We have demonstrated that SIRPα expresses in microglia and regulates immune response as well as phagocytic activity[22]. Despite the work regarding to neuronal CD47, it is still unclear whether microglial SIRPα is actually involved in the negative regulation of synaptic removal. Moreover, the potential role of SIRPα-CD47 axis in synaptopathological diseases such as AD remains to be investigated.

In the present study, we have demonstrated that synaptic density is significantly reduced after microglial SIRPα is ablated in newborn mice, along with excessive synapse pruning by microglia. SIRPα deficiency in microglia undermines its ability to recognize CD47 signal and increases phagocytosis towards synaptic elements. Furthermore, we have provided evidence that microglial SIRPα level declines in the progression of AD. Inhibition of microglial SIRPα signal remarkably exacerbates synaptopathology and cognitive impairment by promoting aberrant synaptic elimination in AD mice model. Taken together, our results suggest that microglial SIRPα negatively regulates synaptic removal, which plays a critical role under both physiological and pathological conditions.

## Result

**Microglial SIRPα specific ablation decreased synaptic density in brain**. First, we assessed SIRPα expression level according to the cell type and developmental stages (P5, P30, and P60). Immunostaining assay revealed that SIRPα primarily expressed in neuron and microglia while there was little expression in astrocyte (Supplementary Fig. 1a). Neuronal SIRPα expression showed a time dependent increase during development stage (Supplementary Fig. 1b, d). Microglial SIRPα level was high at P5, which declined at P30 and maintained this low level at P60 (Supplementary Fig. 1c, e–g). There was no obvious spatial heterogeneity of microglial SIRPα expression in P5 or P30 mice brain (Supplementary Fig. 1h–j). In order to evaluate the contribution of microglial SIRPα in synapse pruning during neurodevelopment, we generated an inducible microglial SIRPα knockout mouse line–Cx3cr1$^{CreERT2}$:SIRPα$^{fl/fl}$ mice (Supplementary Fig. 2a). Mice (Cx3cr1$^{CreERT2}$:SIRPα$^{fl/fl}$ mice) received three consecutive injections of tamoxifen (TAM) once a day from P1 to P3 (Supplementary Fig. 2b). PCR and flow cytometric analysis showed TAM-induced DNA recombination resulted in stable deletion of SIRPα in microglia (CD45$^{int}$ CD11b$^{high}$ cells) (Supplementary Fig. 2c, d).

We applied Cx3cr1$^{CreERT2}$:SIRPα$^{fl/fl}$ mice treated with corn oil as control. SIRPα$^{fl/fl}$ mice receiving TAM were included as another control to rule out potential side-effects caused by tamoxifen. Mice (Cx3cr1$^{CreERT2}$:SIRPα$^{fl/fl}$) treated with TAM were referred as microglial SIRPα specific knockout mice (SIRPα-cKO mice; Fig. 1a).

NeuN staining suggests that the number of neurons and the size of the cell body are not changed in SIRPα-cKO mice (Supplementary Fig. 3a–d, g). Neuronal SIRPα expression was not affected either (Supplementary Fig. 3e, f). Additionally, microglial density and branches were not altered in SIRPα-cKO mice (Supplementary Fig. 4). The expression of SIRPβ1, a closely related family member of SIRPα, was not changed in SIRPα-deficient microglia (Supplementary Fig. 5a–c). To assess the synaptic morphology in SIRPα-cKO mice, we conducted Homer1 and Vglut1 double staining in primary visual cortex (V1) and hippocampus (CA1) at different time points (P5, P15, and P30). Synaptic density was not altered at P5 or P15 in V1 cortex of SIRPα-cKO mice, while it significantly decreased at P30 compared to control (Fig. 1b, d–f and Supplementary Fig. 6a, c, e). The initial synaptic number at P5 or P15 did not alter significantly, indicating that synaptic reduction at P30 is less likely due to early neuron growth defect. Meanwhile, synaptic number in hippocampus of SIRPα-cKO mice showed remarkable decrease at P15 and P30, which occurs earlier than that in primary visual cortex (Fig. 1c, g–i and Supplementary Fig. 6b, d, f). Furthermore, we demonstrated that microglial SIRPα deficiency decreased the frequency, but not the amplitude, of miniature excitatory postsynaptic currents (mEPSCs) in CA1 neuron of P30 mice (Fig. 1j–n), which is coincident with the reduction of synaptic contacts in hippocampus.

Homer1 and Vglut2 double staining revealed that synaptic density was significantly decreased in dLGN of SIRPα-cKO mice at P15 and P30 while these signals remained equivalent at P5 (Supplementary Fig. 7a–d). We further analyzed eye-specific retinogeniculate segregation pattern, which was not affected at P5 but showed significant decrease in overlap between ipsilateral and contralateral signals at P10 and P30, respectively (Supplementary Fig. 7e–h). In addition, SIRPα-cKO and control mice displayed similar innervation of the dLGN in P5 (Supplementary Fig. 7i, j). These data together suggest that microglial SIRPα deficiency decreases synaptic density without affecting neuron growth at early stage (<P5).

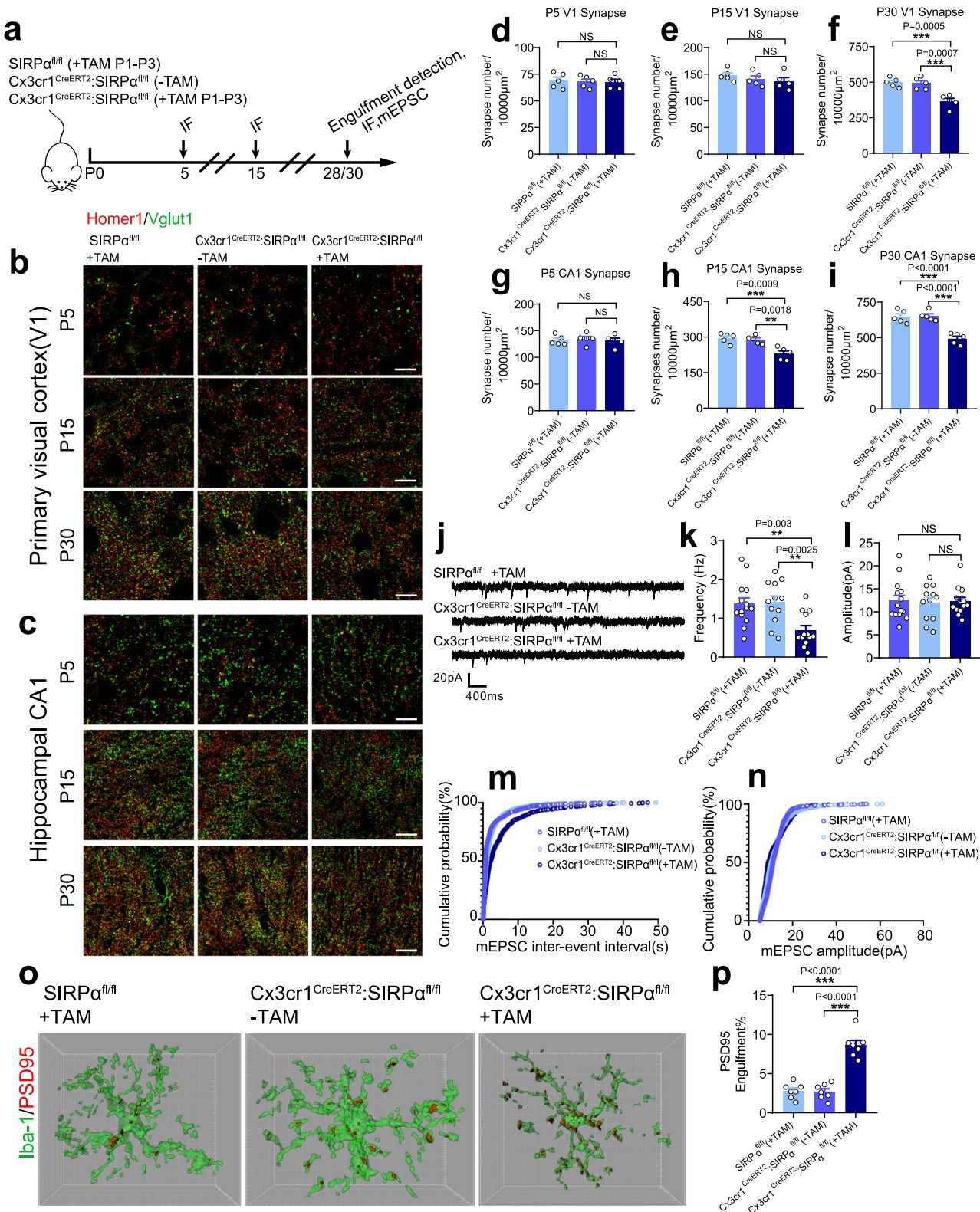

**Increased engulfment of synaptic structure in SIRPα deficient microglia.** To investigate the role of microglia in the synaptic density reduction we observed in SIRPα-cKO mice, we measured microglial engulfment of synaptic elements in cortex (V1) at P28. A negative control that only use secondary antibody demonstrated the specificity of synaptic marker labeling (Supplementary Fig. 8). There was larger volume ratio of PSD95+ or Vglut1+

puncta in Iba-1 positive microglia of SIRPα-cKO mice compared to control (Fig. 1o, p and Supplementary Fig. 6g, h). We also assessed microglial phagocytosis towards synaptic elements in vitro using neuron-microglia co-culture system (Fig. 2a). Iba-1/ MAP2 staining revealed high purify (>90%) of primary microglia/ neurons before coculturing (Supplementary Fig. 9a, b). Both PSD95 and synaptophysin (SPH) staining showed that the overall

**Fig. 1 Synaptic density in microglial SIRPα deficient mice is lower on account of increased microglial engulfment of synaptic structures. a** Schematics of the experimental procedures. IF immunofluorescence, mEPSC miniature excitatory postsynaptic current, TAM tamoxifen, P postnatal. Neonatal mice were injected 50 µg tamoxifen at P1, P2, and P3. **b, c** Representative confocal images depict synaptic staining for presynaptic marker Vglut1 (green) and postsynaptic marker Homer1 (red) in primary visual cortex (V1) (**b**) or hippocampal CA1 stratum radiatum (**c**) in SIRPα$^{fl/fl}$ (+TAM), Cx3cr1$^{CreERT2}$:SIRPα$^{fl/fl}$ (−TAM), and Cx3cr1$^{CreERT2}$:SIRPα$^{fl/fl}$ (+TAM) mice at different time points. Synaptic number was quantified as colocalized pre- and postsynaptic puncta. Scale bar, 10 µm. **d–i** Histograms depict synaptic density in V1 (**d–f**) or hippocampal CA1 (**g–i**) in those mice. $n = 5$ mice/group; average of 10–12 fields from each mouse. One-way ANOVA analysis with Dunnett's multiple comparisons test. **j** Example traces of mEPSCs (miniature excitatory postsynaptic currents) from SIRPα$^{fl/fl}$ (+TAM), Cx3cr1$^{CreERT2}$:SIRPα$^{fl/fl}$ (−TAM), and Cx3cr1$^{CreERT2}$:SIRPα$^{fl/fl}$ (+TAM) mice. **k, l** Histogram shows average mEPSC frequency (**k**) and amplitude (**l**) of SIRPα$^{fl/fl}$ (+TAM), Cx3cr1$^{CreERT2}$:SIRPα$^{fl/fl}$ (−TAM), and Cx3cr1$^{CreERT2}$:SIRPα$^{fl/fl}$ (+TAM) mice. $n = 14, 12, 13$ cells/group from 4 to 5 mice. One-way ANOVA analysis with Dunnett's multiple comparisons test. **m, n** Cumulative probability distributions of mEPSC inter-event intervals (**m**) and amplitude (**n**) of SIRPα$^{fl/fl}$ (+TAM), Cx3cr1$^{CreERT2}$:SIRPα$^{fl/fl}$ (−TAM) and Cx3cr1$^{CreERT2}$:SIRPα$^{fl/fl}$ (+TAM) mice. **o, p** Three-dimensional reconstruction and surface rendering demonstrate larger volumes of PSD95 puncta inside Iba-1 positive microglia in V1 from P28 Cx3cr1$^{CreERT2}$:SIRPα$^{fl/fl}$ (+TAM) mice. $n = 7$ mice/group, average of 8–9 microglia from each mouse, one-way ANOVA analysis with Dunnett's multiple comparisons test. Grid line increments = 5 µm. Data are mean ± s.e.m. **$P < 0.01$, ***$P < 0.001$, NS not significant. Detailed statistical information was listed in Supplementary Statistical Data. Source data are provided as a Source Data file.

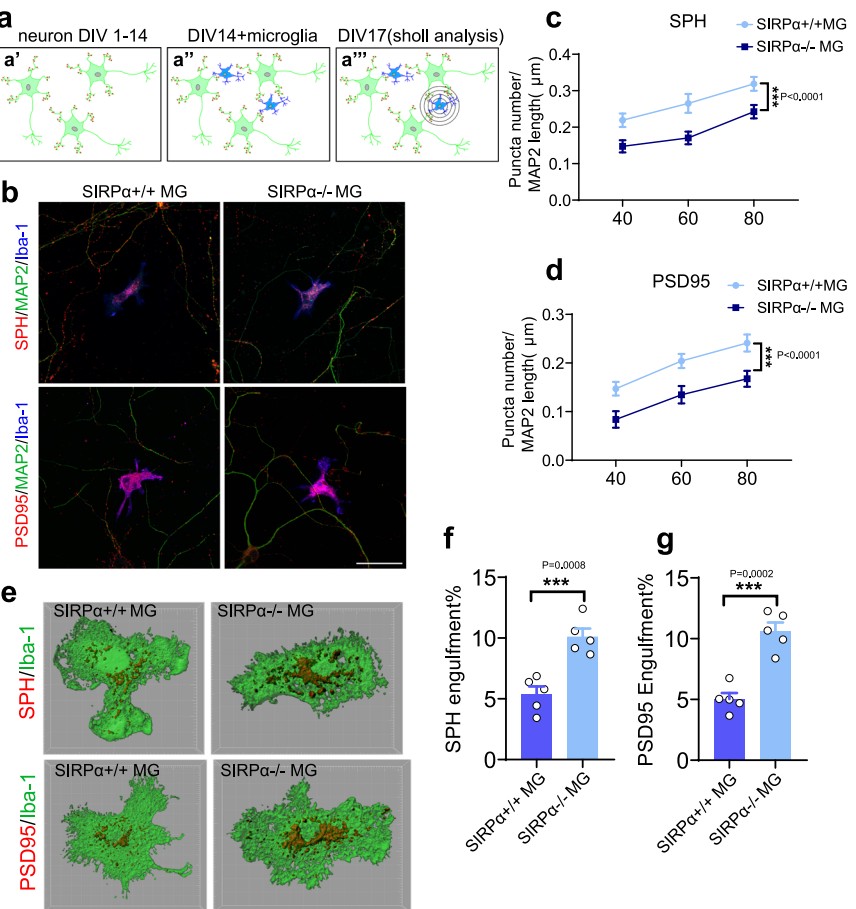

**Fig. 2 Primary cultured SIRPα deficient microglia engulf more synaptic elements. a** Diagrams show neuron-microglia co-cultures (a' and a") and Sholl analyses (a"') that quantifies synaptic puncta around microglia. Green cells in diagrams represent neurons (MAP2+), red puncta represent synaptic marker (SPH/PSD95), and blue cells represent microglia (Iba-1+). **b–d** Confocal images (**b**) show the presence of synaptic puncta (SPH+/PSD95+) around SIRPα+/+ or SIRPα$^{−/−}$ microglia (Iba1+). Scale bar, 50 µm. Statistical analysis for Syanptophysin (**c**) or PSD95 density (**d**) is obtained by two-way ANOVA; $n = 5$ independent experiments. **e–g** 3D reconstruction and surface rendering demonstrate that SIRPα$^{−/−}$ microglia engulf more synaptic elements. Grid line increments = 5 µm. Histograms depict the statistics of synaptic engulfment. Two-tailed unpaired t-test, $n = 5$ independent experiments, average of 3–4 microglia from each experiment. SPH synaptophysin, MG microglia. Data are mean ± s.e.m. ***$P < 0.001$. Detailed statistical information was listed in Supplementary Statistical Data. Source data are provided as a Source Data file.

synaptic density (in area without microglia) was not altered after adding a few microglia with different genotype in the primary neuron culture (Supplementary Fig. 9c, d). However, those PSD95+ or SPH+ synaptic density adjacent to SIRPα$^{−/−}$ microglia (within 80 µm) were significantly reduced compared to control (Fig. 2b–d). In the co-culture system, microglial

morphology (area and circularity) was not significantly altered in SIRPα deficient groups (Supplementary Fig. 9e, f). Most of these microglia (>90%) contained PSD95 or SPH positive structures, which were equivalent in both groups (Supplementary Fig. 9g). We further quantified fluorescent signal of synaptic markers (PSD95 or SPH) inside microglia using 3D reconstruction and

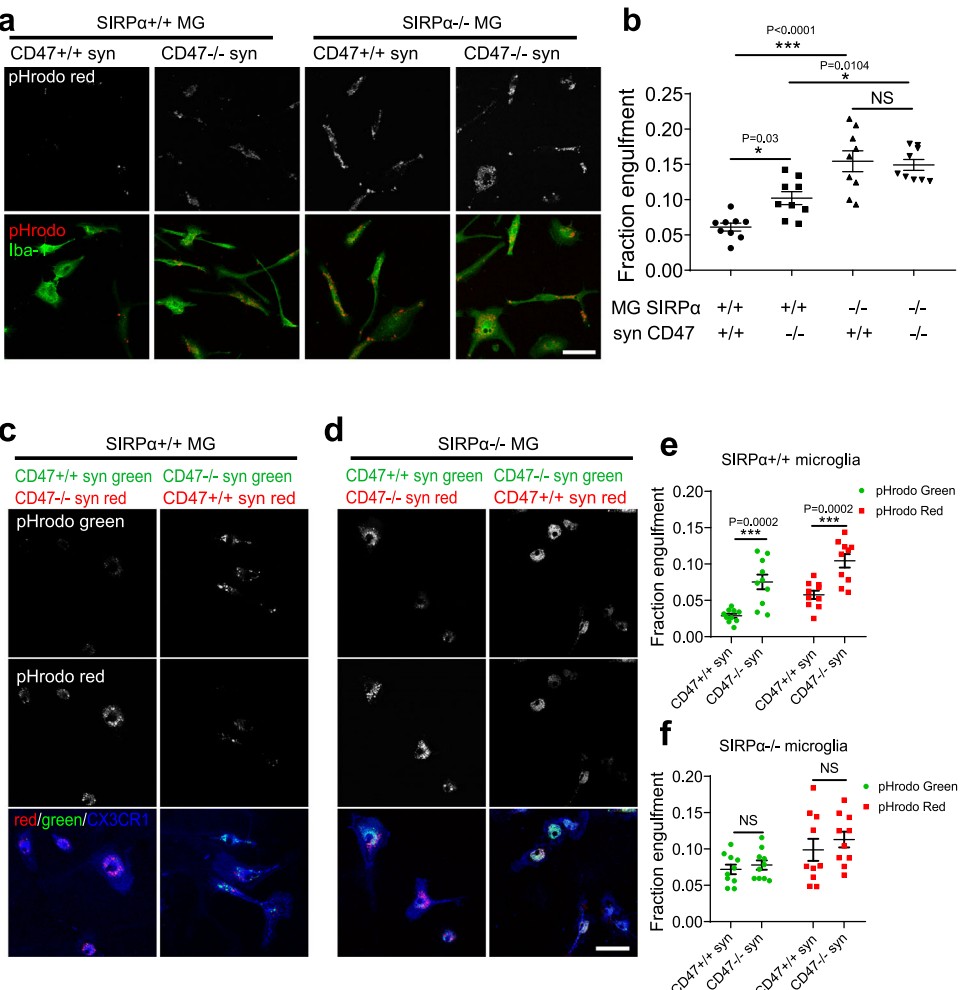

**Fig. 3 SIRPα deficient microglia show increased phagocytic activity. a** Representative images show that primary cultured microglia (SIRPα$^{+/+}$, SIRPα$^{-/-}$) phagocytosed pHrodo red-conjugated synaptosomes (CD47$^{+/+}$, CD47$^{-/-}$). Scale bar, 50 μm. **b** Graph depicts the fraction of synaptosomes engulfed by microglia of different genotypes. $n = 9$ wells; average of 5–6 fields from each well. One-way ANOVA analyses followed by Tukey's multiple comparison test. **c**, **d** Representative images show that CX3CR1 labeled SIRPα$^{+/+}$ (**c**) or SIRPα$^{-/-}$ (**d**) microglia engulfed a mixture of synaptosomes (CD47$^{+/+}$:CD47$^{-/-}$ = 1:1) conjugated with pHrodo-red or pHrodo-green respectively. Scale bar, 50 μm. **e**, **f** Graph depicts the fraction of synaptosomes engulfed by SIRPα$^{+/+}$ (**e**) or SIRPα$^{-/-}$ (**f**) microglia for each combination of pHrodo color. $n = 10$ wells; average of 5–6 fields from each well. Data were analyzed by two-way ANOVA analysis via Sidak's multiple comparisons test. Data are mean ± s.e.m. *$P < 0.05$, ***$P < 0.001$, NS not significant. Detailed statistical information was listed in Supplementary Statistical Data. Source data are provided as a Source Data file.

surface rendering method, and the results revealed more synaptic structures in SIRPα deficient cells (Fig. 2e–g). Additionally, we have demonstrated that microglia engulfed more pHrodo Red-conjugated synaptosomes when SIRPα-CD47 signal is interrupted (Fig. 3a, b). Meanwhile, SIRPα deficient microglia displayed increased phagocytic activity compared to control (Fig. 3a, b).

To further testify whether SIRPα deficiency undermines the ability of microglia to identify CD47 signal, mixture of CD47$^{+/+}$ or CD47$^{-/-}$ synaptosomes labeled with different pHrodo dye were cultured with microglia. Afterwards, we quantified microglial engulfment and demonstrated that SIRPα$^{+/+}$ microglia preferred to engulf CD47$^{-/-}$ synaptosomes (Fig. 3c, e). On the contrary, SIRPα deficient microglia were not able to discriminate the CD47 signal and exhibited indistinguishable phagocytic capacity towards both types of synaptosomes (Fig. 3d, f). These data suggest that SIRPα deficiency in microglia lead to increased phagocytic ability as well as compromised recognition of CD47 signal, which may further result in the excessive elimination of synaptic structures.

Similar results have been achieved in CD47-KO mice. Both morphological and electrophysiological data suggested that synaptic density was reduced in cortex as well as in hippocampus of these mice (Fig. 4a–n; Supplementary Fig. 10). In the meantime, microglia-mediated synaptic clearance during development is remarkably enhanced (Fig. 4o, p). Together, these data indicate that SIRPα-CD47 signal axis protects synapses from excessive elimination mediated by microglia during early neurodevelopment.

**Synaptic CD47 is regulated by neural activity.** Immunostaining results demonstrated that there were both PSD95$^{+}$/CD47$^{+}$ and PSD95$^{+}$/CD47$^{-}$ structures in the neurites of primary neurons (Fig. 5a). Similar observation was obtained when we labeled neurons with CD47 and Synaptophysin (Fig. 5b). We also conducted Vglut1/CD47 or PSD95/CD47 double labeling in brain sections and observed both CD47$^{+}$ and CD47$^{-}$ synaptic elements in vivo (Fig. 5c). These data indicate that CD47 may differentially expressed in neuronal synapses. We further isolated synaptosomes

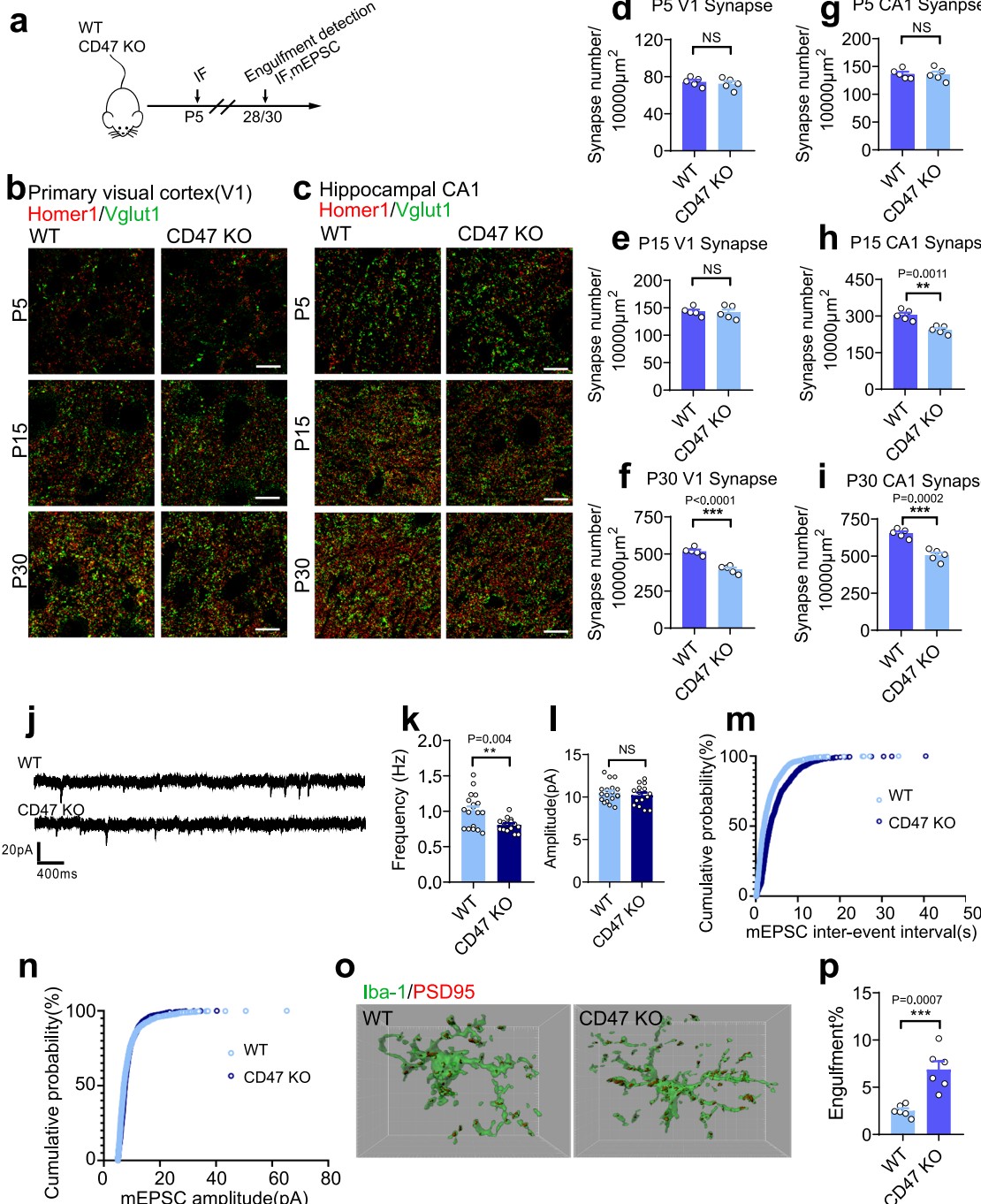

**Fig. 4 Synaptic density in CD47-KO mice is lower on account of enhanced microglial engulfment of synaptic structures. a** Schematics of the experimental procedures. IF immunofluorescence, mEPSC miniature excitatory postsynaptic current. **b, c** Representative confocal images depict synaptic staining for presynaptic marker Vglut1 (green) and postsynaptic marker Homer1 (red) in primary visual cortex (V1) (**b**) and hippocampal CA1 stratum radiatum (**c**) in WT and CD47-KO mice at different time points. Synaptic number was quantified as colocalized pre- and postsynaptic puncta. Scale bars, 10 μm. **d–i** Histograms depict the relative level of synaptic density in V1 (**d–f**) or hippocampal CA1 (**g–i**) in those mice. $n = 5$ mice/group; average of 10–12 fields from each mouse, two-tailed unpaired $t$ test. **j** Example traces of mEPSCs from WT and CD47-KO mice. **k, l** Histogram shows average mEPSC frequency (**k**) and amplitude (**l**) of WT and CD47-KO mice. $n = 18$, 15 cells from 6 to 8 mice, two-tailed unpaired $t$ test. **m, n** Cumulative probability distributions of mEPSC inter-event intervals (**m**) and amplitude (**n**) of WT and CD47-KO mice. **o, p** 3D reconstruction and surface rendering demonstrate larger volumes of PSD95 puncta inside Iba-1 positive microglia in V1 from P28 CD47-KO mice versus WT mice. $n = 6$ mice/group, average of 8–9 microglia from each mouse, two-tailed unpaired $t$ test. Grid line increments = 5 μm. Data are mean ± s.e.m. **$P < 0.01$, ***$P < 0.001$, NS not significant. Detailed statistical information was listed in Supplementary Statistical Data. Source data are provided as a Source Data file.

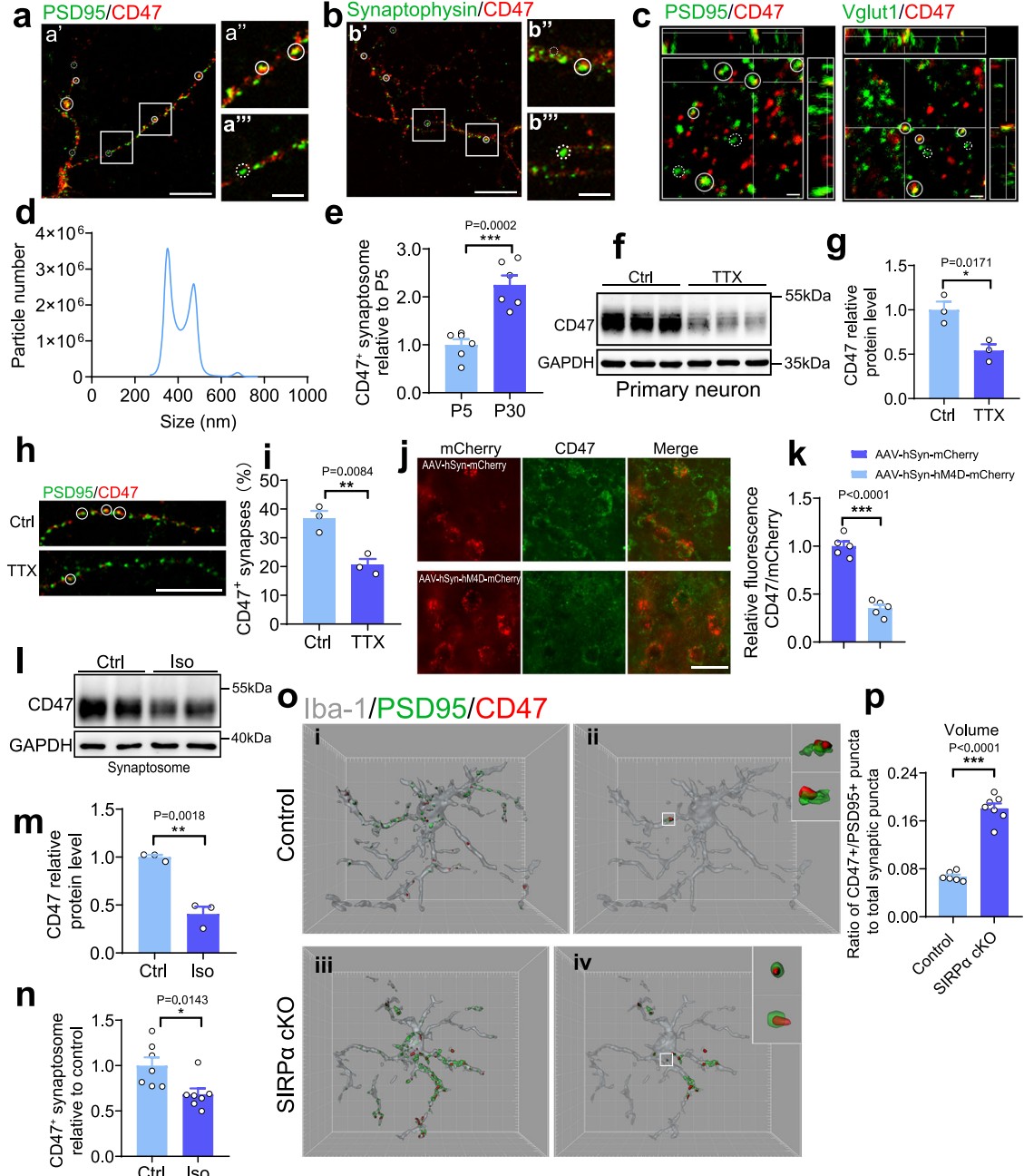

from mice brain and measured CD47 signal by Nanosight. The number of CD47 positive synaptosomes in P30 mice is significantly higher than that of P5 mice (Fig. 5d, e), suggesting an age-dependent increase of CD47 positive synapses in mice brain. As microglia incline to engulf less active inputs during neural refinement, we hypothesized that low level of neural activity may decrease CD47 signal, which facilitates microglia-mediated synaptic elimination. To assess impact of neural activity on CD47 expression, channel-blocker tetrodotoxin (TTX) was applied and it significantly decreased level of CD47 expression as well as CD47 positive synapses in primary neuron (Fig. 5f–i). We also overexpressed hM4D (Gi) with mCherry in cortical neurons in adult mice (Supplementary Fig. 11). After inhibiting the neuron activity by CNO (clozapine-N-oxide) administration, we measured the protein level of CD47 in those neurons and found remarkable decrease of CD47 expression compared to control (Fig. 5j, k). Synaptosomes isolated from general anesthetic (isoflurane treated) mice brain showed reduced CD47 expression compared to control

(Fig. 5l, m). Nanosight analysis suggested that number of CD47 positive synaptosomes in anesthetic mice is downregulated as well (Fig. 5n). Moreover, phagocytic assay revealed that microglia preferred to engulf those synaptosomes derived from anesthetic mice brain, indicating CD47 downregulation facilitates microglia-mediated synaptic clearance (Supplementary Fig. 12a, b).

We further compared amount of CD47[+]/PSD95[+] puncta in microglia from SIRPα-cKO and control mice. In control microglia, only a few of PSD95[+] puncta were CD47 positive, indicating the preferential engulfment of CD47 negative synapses. In SIRPα deficient microglia, ratio of CD47[+]/PSD95[+] puncta to total PSD95[+] puncta were significantly increased, suggesting that the selective engulfment of CD47[−] synapses was abolished in microglia after losing SIRPα (Fig. 5o, p). Furthermore, we examined ratio of CD47[+]/PSD95[+] synaptic structure in microglia from mice under anesthesia condition. Isoflurane treatment increased microglial engulfment of PSD95[+] synaptic structures (Supplementary Fig. 12c, d). Most of those PSD95[+] elements

**Fig. 5 CD47 is expressed in synapses and regulated by neural activity. a, b** Confocal images show CD47 co-expression with presynaptic marker (synaptophysin, SPH) or postsynaptic marker (PSD95) in primary neurons. (a″) CD47$^+$/PSD95$^+$ puncta are marked by circle; (a‴) CD47$^-$/PSD95$^+$ puncta are marked by dashed circle; (b′) CD47$^+$/SPH95$^+$ puncta are marked by circle; (b″) CD47$^+$/SPH95$^+$ puncta are marked by dashed circle; Scale bars, (a′, b′) 20 μm; (a‴, b″) 5 μm. This experiment was repeated independently for three times. **c** Orthogonal views of CD47/PSD95 (postsynaptic marker) or CD47/Vglut1 (presynaptic marker) double staining in brain section in P28 mice, CD47$^+$ or CD47$^-$ synaptic elements were marked by circleor dashed circle. Scale bar, 1 μm. This experiment was repeated independently for three times. **d** Nanosight shows size range of synaptosomes passed through membrane filters (pore size 0.45 μm). **e** Synaptosomes conjugated with CD47-FITC antibody are quantified by Nanosight and the relative level of CD47$^+$ synaptosomes in adult mice is significantly increased compared to that in newborn mice. $n = 6$ mice/group, two-tailed unpaired $t$ test. **f, g** Western blot analysis of primary neurons shows that neuronal CD47 expression was downregulated after TTX treatment (1 μM for 48 h). $n = 3$ independent experiments, two-tailed unpaired $t$ test. **h, i** Representative images and statistical analysis show the percentage of CD47$^+$ synaptic structures (CD47$^+$PSD95 puncta/total PSD95 puncta) was reduced in TTX (tetrodotoxin) treated neurons. Scale bar, 20 μm. $n = 3$ independent experiments, average 6–7 fields from each assay; two-tailed unpaired $t$ test. **j, k** Representative images and statistical analysis demonstrate that CD47 level decreased in neuron expressing the inhibitory DREADD-hM4Di in response to CNO stimulation. AAV-hSyn-hM4D(Gi)-mCherry or control virus (100 nL, $1.25 \times 10^{13}$ GC/mL) was injected into mouse primary visual cortex, 4 weeks later mice were subjected to Clozapine-N-oxide (CNO) administration for 3 consecutive days. Scale bar, 20 μm; $n = 5$ mice/group, average of 8–10 fields from each mouse, two-tailed unpaired $t$ test. **l, m** Western blot analysis of synaptosomes isolated from isoflurane (iso) treated mice also shows reduction of CD47 compared to control mice, $n = 3$ mice/group, two-tailed unpaired $t$ test. **n** Nanosight detection reveals that relative level of CD47$^+$ synaptosomes isolated from isoflurane (iso) treated mice is significantly reduced. $n = 7$ mice/group, unpaired $t$ test. **o** Left panels show CD47$^+$ or PSD95$^+$ signal inside microglia from SIRPα-cKO and control mice (i and iii); right panels (ii and iv) show CD47 and PSD95 colocalized puncta inside cells. Insets are enlarged images with orthogonal plane of typical CD47$^+$/PSD95$^+$ puncta. Grid line increments = 5 μm. **p** Histogram shows volume ratio of CD47$^+$/PSD95$^+$ puncta to total synaptic structures, $n = 6, 7$ mice/group, average of 8–9 cell from each mouse, two-tailed unpaired $t$ test. Data are mean ± s.e.m. *$P < 0.05$, **$P < 0.01$, ***$P < 0.001$, NS not significant. Detailed statistical information was listed in Supplementary Statistical Data. Source data are provided as a Source Data file.

were CD47 negative, ratio of CD47$^+$/PSD95$^+$ puncta to total synaptic structures showed a little but significant declination compared to mice under normal condition (Supplementary Fig. 12e). Together, these data indicate that CD47 expression may decrease in less active synapses which are prone to be eliminated by microglia.

**Microglial SIRPα is downregulated in AD pathology**. We further investigated the role of microglial SIRPα in AD, in which loss of synapses is a central neuroanatomical hallmark. The protein level of SIRPα was remarkably decreased in the cortex of AD patients (Fig. 6a, b). It was also downregulated in AD mice brain (6 months and 12 months old) compared to same-aged control (Supplementary Fig. 13a, b). In the meantime, SIRPβ1 expression is elevated in both human AD patients and AD mice brain (Supplementary Fig. 5d–h).

To specifically assess the alteration of SIRPα in microglia during AD, we conducted flow cytometric analysis and showed that microglial SIRPα did not change significantly in WT mice at different time points (2, 5, and 8 months). On the contrary, SIRPα started to decline in microglia of AD mice at 5 months age (before plaque formation) and continued to decrease at 8 months old (after plaque formation; Fig. 6c, d). Immunostaining assay confirmed the declination of microglial SIRPα in 5- and 8-months-old AD mouse brain, while microglial SIRPα levels were similar between AD and WT mice at 2 months age (Fig. 6e, f and Supplementary Fig. 13c–f). When we challenged primary microglia with soluble Aβ$_{42}$ oligomer (Aβo), the protein level of SIRPα significantly decreased (Fig. 6g, h). Flow cytometric analysis also revealed marked reduction of microglial SIRPα in mice brain followed with intracerebroventricular (ICV) injections of Aβo (Fig. 6i, j). These data suggest that Aβ oligomer generated in the early stage of AD inhibits SIRPα expression in microglia.

**SIRPα deficiency enhanced cognitive impairment and synaptic loss without modifying Aβ plaque depositions**. To exclude the potential impact of neuronal SIRPα as well as the developmental change caused by early microglial SIRPα deletion, we have applied AD$^{APPswe, PSEN1dE9}$/SIRPα-cKO mice model (by tamoxifen induction, microglial SIRPα was specifically ablated at

2 months age) to investigate the role of microglial SIRPα deficiency on AD pathogenesis in adulthood (Fig. 7a). Open field analysis showed similar locomotor activity among all those mice groups (control; SIRPα-cKO; AD; AD/SIRPα-cKO) at 5 months age (Fig. 7b). Although we observed decreased locomotor activity in mice with AD background at 8 months age, there were no significant differences between AD and AD/SIRPα-cKO mice (Fig. 7b). In addition, all groups of mice showed similar escape latencies in the visible platform test (Fig. 7c), which implies comparable vision among these mice. Hidden platform test and spatial probe trial results suggested that microglial SIRPα ablation accelerated cognitive impairment in AD mice at 5 months age, even preceding the plaque formation (Fig. 7d, e). Although such effect persisted in 8-months-old mice (Fig. 7d, e), we did not find any significant variations in amyloid deposition between AD and AD/SIRPα-cKO mice at this stage (Fig. 7f, g). ELISA analysis also demonstrated that both soluble and insoluble Aβ$_{42}$ were not significantly changed between these mice at 5- or 8-months age (Fig. 7h, i). Iba-1 and 6E10 double staining revealed that microglial recruitment around plaques between AD and AD/SIRPα-cKO mice were indistinguishable (Fig. 7j, k). These data suggest that microglial SIRPα deficiency has little impact on amyloid pathology in AD mice.

PSD95/Synapsin I labeling showed that microglial SIRPα deficiency remarkably enhanced synaptic loss in cortex and hippocampus of AD mice at 5- and 8-months age (Fig. 7l, m and Supplementary Fig. 14). Additionally, Golgi staining demonstrated consistent results that dendritic spine density was less in AD/SIRPα-cKO mice (Fig. 7n, o). Notably, ablation of microglial SIRPα at 2-months-age did not alter synaptic number in these mice with non-AD background at 5 months or 8 months age (Fig. 7l–o and Supplementary Fig. 14). These data suggest that microglial SIRPα deficiency in adulthood accelerated synapse loss and cognitive declination in AD mice.

**Loss of microglial SIRPα increased Aβo-induced synapse loss by promoting microglia-mediated synaptic elimination**. We have provided evidence that Aβo downregulated microglial SIRPα expression. To further specify its role in synaptopathology during AD progression, we applied another AD model using SIRPα-cKO mice followed with ICV injection of Aβo (Fig. 8a). Specific gene

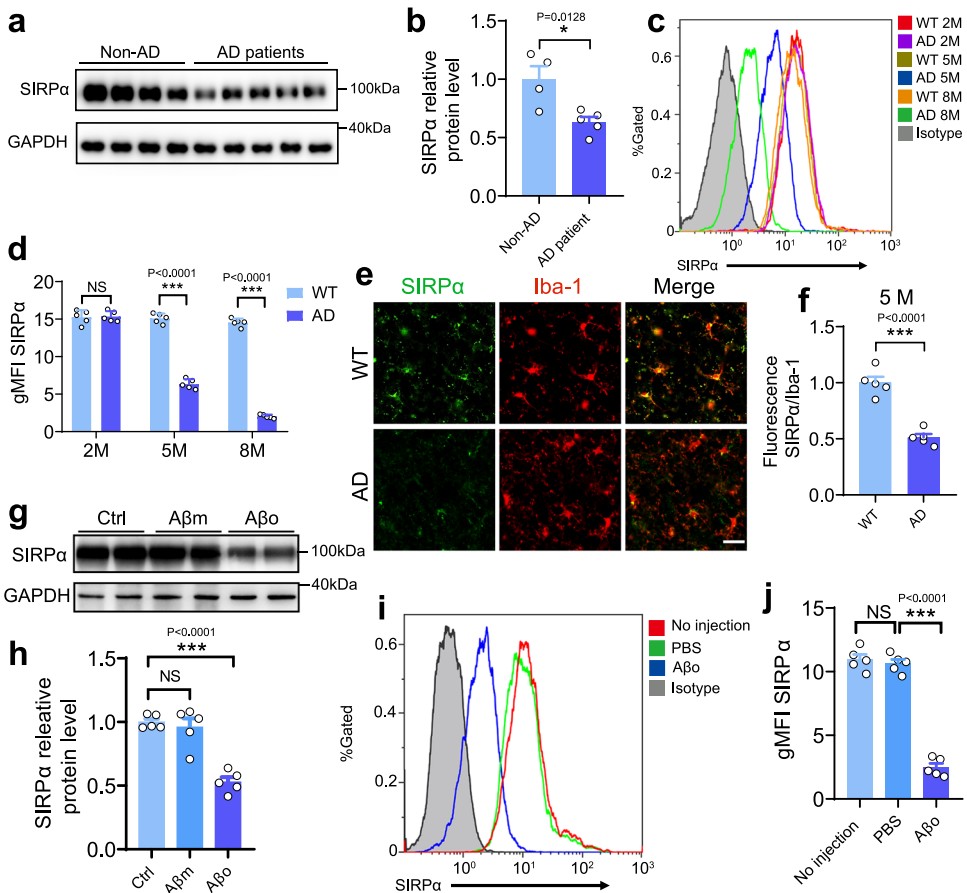

**Fig. 6 Microglial SIRPα is downregulated in AD pathology. a**, **b** Western blot and statistical analysis show SIRPα expression decreased remarkably in the cortex of AD patients. Non-AD = 4, AD patients=5. Two-tailed unpaired *t* test. Detailed sample information is listed in the Table 1 in Methods section. **c**, **d** Flow cytometry analysis displays SIRPα expression of acute isolated microglia from 2-months, 5-months, and 8-months old WT and AD mice. Histograms represent quantification of geometric mean fluorescent intensity (gMFI) of SIRPα. *n* = 5 mice/group, average of three tests from each mouse, two-way ANOVA via Sidak's multiple comparisons test. **e**, **f** Immunostaining and quantification of SIRPα with microglial marker (Iba-1) in cortex of WT and AD mice (5 months old). Scale bar, 25 μm, *n* = 5 mice/group; average of 6–8 fields from each mouse, two-tailed unpaired *t* test. **g**, **h** Western blot and statistical analysis of SIRPα protein level in primary cultured microglia after Aβ42 monomer (Aβm) or Aβ42 oligomer (Aβo) treatment. Cells were treated with 0.2 μM Aβ for 24 h before protein analysis. *n* = 5 experiments, one-way ANOVA, Dunnett's multiple comparisons test. **i**, **j** Flow cytometry analysis shows microglial SIRPα decreased 3 days after intracerebroventricular (ICV) injection of Aβo (2 μg/mouse). Histogram shows the gMFI quantification of microglial SIRPα. *n* = 5 mice/group (WT male, 3 months age), average of three tests from each mouse. One-way ANOVA analysis with Dunnett's multiple comparisons test. Data are mean ± s.e.m. *P < 0.05, ***P < 0.001, NS not significant. Detailed statistical information was listed in Supplementary Statistical Data. Source data are provided as a Source Data file.

deletion is induced by TAM injection in 2-months-old mice to exclude potential impact of microglial SIRPα deficiency during early neurodevelopment (Fig. 8a). After Aβo injection (3-months-old mice), synaptic number significantly reduced in SIRPα-cKO mice compared to that in control (Fig. 8b, c), which is consistent to results we achieved in AD/SIRPα-cKO mice. Similarly, deletion of microglial SIRPα at 2-months-age had little impact on synapses of those mice without Aβo stimulation (Fig. 8b, c).

We further examined microglial engulfment towards synaptic elements and demonstrated that there was significantly larger volume of PSD95+ structures in microglia of SIRPα-cKO mice after Aβo challenge (Fig. 8d, e). Besides, flow cytometric analysis detected more microglia with PSD95 positive elements in SIRPα-cKO mice (Fig. 8f, g). These data suggest that phagocytic activity of microglia towards synaptic structures is increased in SIRPα-cKO mice upon Aβ stimulation, leading to enhanced synaptic loss during AD pathology. In vitro assay revealed that both Aβm and Aβo stimulation increased phagocytosis of synaptosomes in primary microglia (Supplementary Fig. 15a, b), which may due to microglia activation. However, Aβo treatment decreased

microglial SIRPα expression (Fig. 6g, h), which was associated with greatly enhanced phagocytosis compared to Aβm-treated group (Supplementary Fig. 15a, b). This result suggests that downregulation of SIRPα after Aβo treatment further increases phagocytosis of synaptosomes in microglia. In addition, over-expressing SIRPα via lentivirus infection remarkably reduced microglial engulfment towards synaptosomes after Aβo treatment (Supplementary Fig. 15c, d).

To further elucidate the role of SIRPα-CD47 signal axis in AD pathology, we examined CD47 protein level in primary neuron subjected to Aβ stimulation and found that Aβo treatment significantly decreased CD47 expression (Supplementary Fig. 16a, b). Synaptosome CD47 level was also declined in AD mouse brain at 8-months age (Supplementary Fig. 16c, d). Those AD mice (8 M) derived synaptosomes were easier to be engulfed by microglia (Supplementary Fig. 16e, f). In addition, we performed ICV injection of Aβo in CD47-KO mice and found those mice were more vulnerable upon Aβo stimulation, displaying greater loss of synapses and enhanced microglial engulfment compared to control (Supplementary Fig. 16g–k). Together, these data

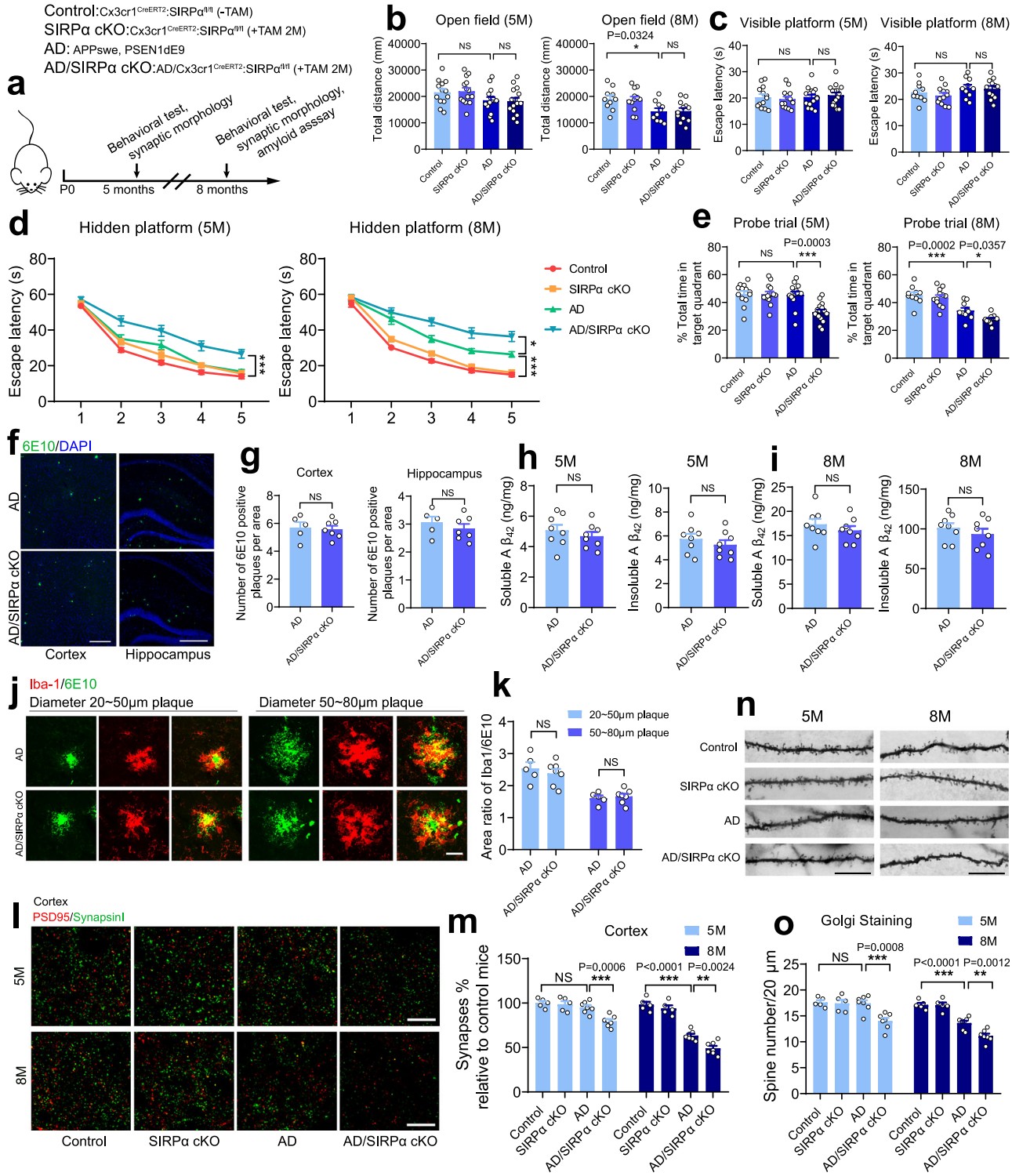

indicate that dysregulation of SIRPα-CD47 signal axis is involved in the inappropriate synaptic elimination in AD pathology.

## Discussion

Microglia are of great significance in synaptic remodeling in central nervous system while the molecular mechanism under which is still under investigation[23,24]. Several studies have demonstrated that iC3b/C3b-CR3 signal positively regulates the removal of synapse during neurodevelopment[3,25,26]. Besides, there is evidence that neuronal CD47 can protect synapse from

inappropriate elimination by microglia in visual system during early development–the first investigation of negative regulator in synaptic pruning in CNS, in which they considered SIRPα-CD47 signal axis plays an important role[9]. However, the exact effects of microglial SIRPα on synaptic remodeling in CNS remain obscure. Here we have demonstrated that microglial SIRPα-cKO mice have reduced synaptic number not only in dLGN but also in visual cortex as well as in hippocampus. Similar results were achieved in CD47-KO mice, indicating SIRPα-CD47 signal may act generally in negative regulation of synaptic remodeling in brain. We have provided evidence that microglia engulfed more

**Fig. 7 SIRPα deficiency enhanced cognitive impairment and synaptic loss without modifying Aβ plaque depositions. a** Schematics of the experimental procedures, Cx3cr1[CreERT2]:SIRPα[fl/fl] (-TAM) mice are used as normal control. TAM tamoxifen. **b** Histograms depict that total distance mice traveled in open filed experiment has no significant difference. 5 M: $n = 13, 14, 15, 15$ mice/group; 8M: $n = 10, 12, 10, 12$ mice/group, one-way ANOVA, by Dunnett's multiple comparisons test. **c** Histograms depict escape latency of four different groups of mice in visible platform test (time to find the visible platform) before plaque deposition (5 months old) and after plaque deposition (8 months old). 5M: $n = 13, 14, 15, 15$ mice/group; 8 M: $n = 10, 12, 10, 12$ mice/group, one-way ANOVA, by Dunnett's multiple comparisons test. **d** Line charts show Morris Water Maze escape latency (time to find the hidden platform) in five consecutive days of four different genotypic mice at the age of 5 months old and 8 months old respectively. 5M: $n = 13, 14, 15, 15$ mice/group; 8M: $n = 10, 12, 10, 12$ mice/group, two-way RM ANOVA, via Tukey's multiple comparisons test. **e** Histograms depict the time spent in target quadrant in Morris Water Maze probe trial of four different genotypic mice at the age of 5 months old or 8 months old. 5M: $n = 13, 14, 15, 15$ mice/group, Kruskal–Wallis test via Dunn's multiple comparisons test; 8M: $n = 10, 12, 10, 12$ mice/group, one-way ANOVA via Dunnett's multiple comparisons test. **f, g** Representative immunostaining images exhibit 6E10 positive plaques in cortex and hippocampus of 8-months-old AD and AD/SIRPα-cKO mice. Scale bar, 200 μm. Quantification analyses show that 6E10 positive plaques in AD and AD/SIRPα-cKO mice were equivalent. $n = 5, 7$ mice/group; average of 5–6 fields from each mouse, two-tailed unpaired $t$-test. **h, i** ELISA analyses of both soluble and insoluble Aβ$_{42}$ in 5-months-old and 8-months-old AD and AD/SIRPα-cKO mice brain. $n = 8$ mice/group, two-tailed unpaired $t$-test. **j, k** Representative images of 6E10 immunoreactive plaques (diameter 20–50 μm as small groups; 50–80 μm as big group) surrounded by microglia in 8-months-old AD and AD/SIRPα KO mice. Scale bar, 25 μm. Histograms show the area ratio of Iba-1/6E10 are comparable in AD and AD/SIRPα-cKO mice. $n = 5, 7$ mice/group. Average of 3–4 plaques from each mouse, two-tailed unpaired $t$-test. **l, m** Representative confocal images depict synaptic staining for presynaptic marker Synapsin I (green) and postsynaptic marker PSD95 (red) in cortex of 5-months-old and 8-months-old mice. Synaptic number was determined as colocalized pre- and postsynaptic puncta. Scale bars, 10 μm. $n = 5, 5, 7, 7$ mice/group, average of 10–12 fields from each mouse, one-way ANOVA via Dunnett's multiple comparisons test. **n, o** Golgi staining images of apical dendritic spine in cortex of 5-months-old and 8-months-old mice. Scale bars, 20 μm. $n = 5, 5, 7, 7$ mice/group, average of 10 dendrites from each mouse; one-way ANOVA via Dunnett's multiple comparisons test. Data are mean ± s.e.m. *$P < 0.05$, **$P < 0.01$, ***$P < 0.001$, NS not significant. Detailed statistical information was listed in Supplementary Statistical Data. Source data are provided as a Source Data file.

synaptic structure in SIRPα cKO and CD47-KO mice during development. Besides, synaptic number was not altered in these mice at early stage of development (P5), indicating that microglia-mediated over-pruning may be involved in the synaptic reduction we observed. However, based on the results that number of synapses increases from P15 to P30, we cannot rule out the possibility that deficiency of SIRPα-CD47 signal affects synaptic formation, which may also contribute to the synaptic reduction at this duration.

It is reported that the Cx3cr1[CreERT2] strain exhibits considerable leakiness. Van Hove et al.[27] analyzed tamoxifen-independent CreERT2 recombination, demonstrating <3% leaky expression when the length of the floxed sequence is >2.5 kb. In our SIRPα-cKO model, the genetic distance between 2 loxP sites is 14.4 kb, which shows little leakage effect without TAM. Deletion of SIRPα in microglia impacts its phagocytic ability, which leads to the excessive elimination of synaptic structures in SIRPα-cKO mice brain during early development. The phagocytic ability of microglia is not only the capacity to remove the functional-less waste but also the ability to recognize them. Maintaining brain homeostasis during development depends on the coordination of these two aspects[28,29]. Our data showed that SIRPα deficiency increased microglial phagocytosis towards synaptic structures. When co-incubated with mixed synaptosomes with/without CD47 expression, wild-type microglia preferred to engulf CD47 negative synaptosomes. In the meantime, SIRPα-deficient microglia exhibited impaired ability in discriminating CD47 signal by selecting both types of synaptosomes with no differences. In SIRPα-cKO mice, microglia engulfed more synaptic structures during early neurodevelopment. There was higher ratio of CD47[+]/PSD95[+] synaptic elements in SIRPα-cKO microglia compared to control, indicating these microglia failed to recognize CD47 signal (do not eat me signal) after losing SIRPα. Although microglial engulfment towards synapses in anesthetic mice increased as well, the ratio of CD47[+]/PSD95[+] synaptic elements were decreased. This may be the result of decreased synaptic CD47 expression after neural activity inhibition, leaving more CD47 negative synaptic structures to be eliminated by microglia. These data indicate that the presence of SIRPα enables microglia to preserve synaptic structures by recognizing CD47 signal.

We have provided evidence that CD47 expresses differentially in synapses/synaptosomes. Inhibiting neural activity significantly decreases neuronal CD47 expression both in vitro and in vivo. Additionally, isoflurane treatment resulted in remarkable reduction of CD47 level in synaptosomes isolated from newborn mice, which is consistent with the previous report that CD47 is regulated by neural activity. Those isolated synaptosomes were easier to be engulfed by microglia, indicating that those less active synapses may be eliminated by microglia due to the loss of SIRPα-CD47 signal[9]. Interestingly, Toth et al.[30] have observed that synaptic density is decreased in hippocampus of SIRPα KO mice (generated by crossing SIRPα floxP mice with Actin-Cre line), in which they considered it as a result of inhibited synapse maturation due to the loss of neuronal SIRPα. Here, we provide evidence that loss of microglial SIRPα may also contribute to the reduced synaptic density during early neurodevelopment. We have demonstrated that microglial SIRPα deletion in newborn mice increased synaptic elimination during pruning period, which showed persistent effects even in P30 mice. In our AD study, we induced microglial SIRPα ablation in two-months old mice (after the peak of pruning period) to exclude the potential impact of early SIRPα deficiency. Adult microglial SIRPα deficiency can no longer affect early synaptic pruning and shows little impact on synaptic density at later stages in normal condition. Consistently, those SIRPα-cKO mice did not show any behavioral changes as well. It seems that SIRPα-CD47 axis, as the negative regulatory signal, modulates the synaptic remodeling in a passive manner during neurodevelopment. SIRPα deficiency just releases the brake of microglia, it takes effect only when the synaptic elimination process is initiated.

In addition to the physiological conditions, positive signals that modulate microglia-mediated synaptic removal also play critical roles in some disease-related neuropsychiatric symptoms. For instance, C1q level is aberrantly upregulated in neuronal synapses of AD model, which resulted in microglia-mediated early synaptic loss[18]. Blocking the C1q-dependent iC3b/C3b-CR3 signaling pathway can significantly alleviate memory loss in AD mice[15]. Similar complementary pathways are also activated in West Nile Virus infection, lupus or schizophrenia, which may further lead to dysregulated synaptic removal and significant neurological symptoms[16,20,31]. Besides, abnormal microglia derived from

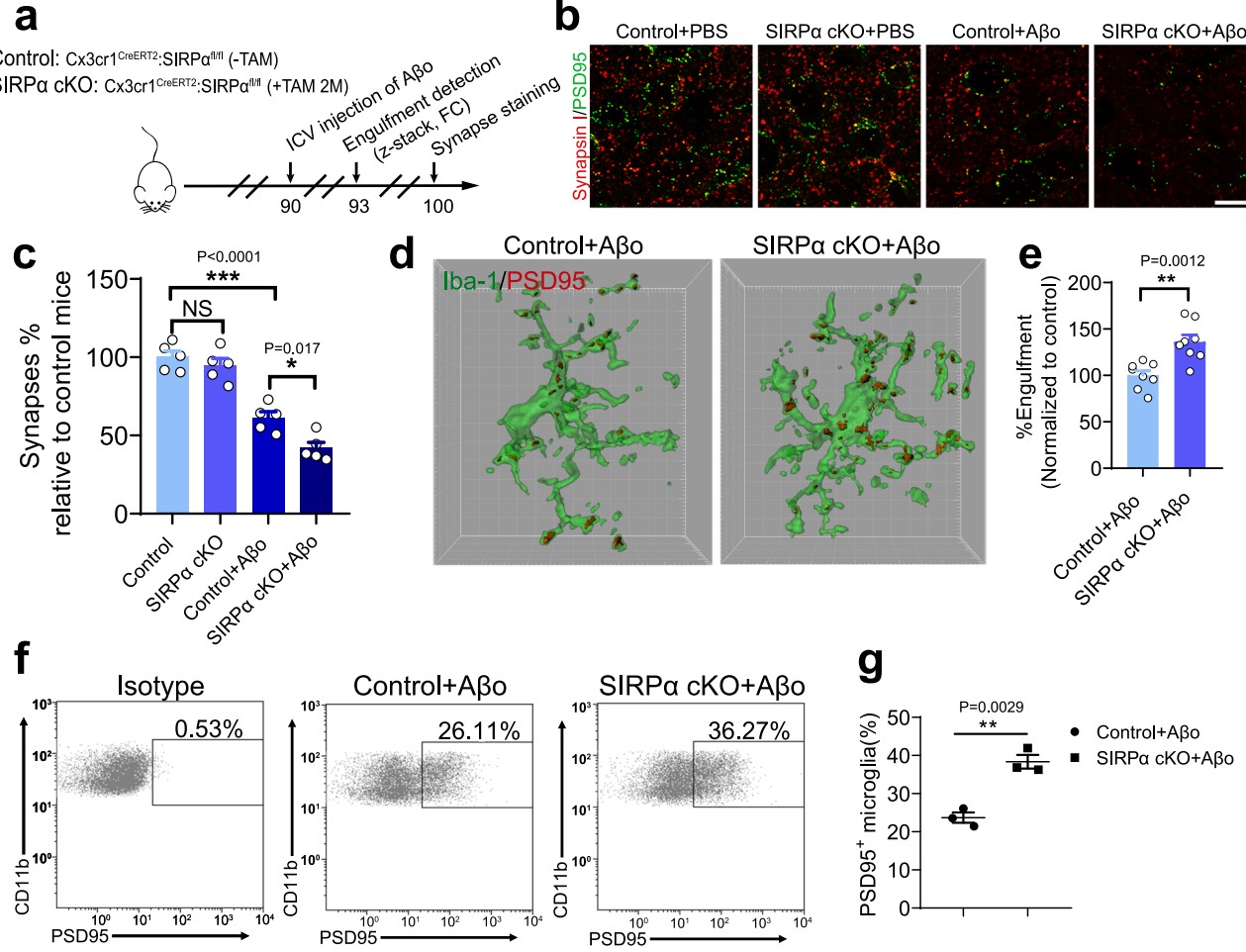

**Fig. 8 Inhibition of microglial SIRPα increased Aβo-induced synapse loss by promoting microglia-mediated synaptic elimination. a** Schematics of the experimental procedures, Cx3cr1[CreERT2]:SIRPα[fl/fl] (-TAM) mice are used as normal control. ICV intracerebroventricular, FC flow cytometry, TAM tamoxifen. **b, c** Representative confocal images (**b**) depict synaptic staining for presynaptic marker Synapsin I (red) and postsynaptic marker PSD95 (green) in cortex, Scale bars, 10 μm. Aβm, Aβ$_{42}$ monomer; Aβo, Aβ$_{42}$ oligomer. The histogram **c** displays the quantification of synaptic density in these mice. $n = 5$ mice/group (male, 3 months age), average of 6–8 fields from each mouse. One-way ANOVA, by Tukey's multiple comparison test. **d, e** 3D reconstruction and surface rendering demonstrate larger volumes of PSD95[+] puncta inside Iba-1[+] microglia in cortex from SIRPα-cKO mice versus control mice after Aβo stimulation (2 μg/mouse), Grid line increments = 5 μm. $n = 8$ mice/group, average of 8–9 microglia from each mouse, two-tailed unpaired $t$ test. **f, g** Flow cytometry analysis of synaptic material (PSD95[+]) engulfed by microglia in SIRPα-cKO or control mice after Aβo injection. $n = 3$ mice/ group, average of three tests from each mouse, two-tailed unpaired $t$ test. Data are mean ± s.e.m. *$P < 0.05$, **$P < 0.01$, ***$P < 0.001$, NS not significant. Detailed statistical information was listed in Supplementary Statistical Data. Source data are provided as a Source Data file.

schizophrenia patient showed increased phagocytosis of synaptic structure, indicating microglial dysfunction contributes to the aberrant synaptic elimination during these disease progression[6]. Nevertheless, the role of negative regulatory signals of synaptic elimination in neurological disorder is largely unknown. In the present study, we have demonstrated that microglial SIRPα expression is downregulated during AD progression while SIRPβ1, which shares similar extramembrane domain with SIRPα but has a very different intracellular domain, is elevated both in human AD patients and AD mice brain[32,33]. All these changes indicate the increased phagocytic activity of microglia in AD[32]. While SIRPβ1 induces phagocytosis towards cell debris and Aβ that is beneficial to the disease progression, our results show that downregulation of SIRPα enhances phagocytic activity towards synaptic elements that has detrimental effects. Whether microglial phagocytosis has double-edged effects or they are separately under control by different signal pathway needs further investigation. Downregulation of microglial SIRPα may result from the inhibition by Aβo generated in the early stage of the disease. Aβo

can induce pro-inflammatory phenotype of microglia and there is evidence that microglial SIRPα is reduced during aging or inflammatory stress, indicating that chronic inflammation may contribute to Aβo-induced SIRPα reduction in microglia[22]. These data together suggest the involvement of microglial SIRPα signal in AD as well as other aging related neurodegenerative diseases.

Using AD/SIRPα-cKO mice in which microglial SIRPα was specifically ablated at 2 months age, we can exclude the potential impact of neuronal SIRPα deficiency as well as the developmental change caused by early microglial SIRPα deletion. We found microglial SIRPα deficiency had little effect on the amyloid deposition, while it increased synapses loss and enhanced memory impairment in AD mice. Although phagocytic capability of SIRPα deficient microglia is enhanced, the ability of target recognition is compromised due to the blockage of SIRPα-CD47 signal. Moreover, the recruitment of microglia around the plaque was comparable between AD/SIRPα-cKO mice and AD control. Therefore, the overall effects of SIRPα deficiency on amyloid deposition are insignificant. We observed reduced synaptic

density in SIRPα-cKO mice followed with Aβo administration. 3D imaging revealed that SIRPα deficient microglia engulfed more synaptic structures in Aβ mice brain. In these models, Aβ acted as an initial insult that triggers neurotoxic cascade. In addition, it also induces microglia-mediated synaptic loss by activating C1q-dependent iC3b/C3b-CR3 signaling pathway[18]. Our results suggest that lack of microglial SIRPα signal may further exacerbate the situation. Notably, ablation of microglial SIRPα at 2 months old shows little impact on synaptic density in non-AD mice while it induces excessive synaptic elimination in AD mice at later stages. These data suggest SIRPα signal modulates the synaptic removal in a passive manner when synaptic elimination process is initiated by Aβ during AD pathology. Overexpression of SIRPα in Aβ stimulated microglia can prevent its excessive phagocytosis towards synaptosomes. Indeed, these results would indicate that SIRPα-CD47 signal axis may be a potential target in synaptopathology of AD.

CD47 expression is also downregulated during AD progression. In AD mice, synaptosomal CD47 level decreases at 8 months age. The declination of CD47 expression at this stage may be the results of long-term decreased neural activity[34], as in anesthetized mice, we also observed CD47 downregulation in synaptic structure. Notably, there was evidence that microglial surveillance is significantly increased in anesthetized mice model, indicating the potential increase of microglia-mediated synaptic clearance[35,36]. Furthermore, synaptosomes isolated from late stage of AD mice brain (8 M) with lower CD47 expression are easier to be engulfed by microglia. CD47-KO mice phenocopy the enhanced loss of synapses and microglial engulfment in SIRPα-cKO mice followed with ICV injection of Aβo, suggesting that SIRPα and CD47 may work together in the synaptic pathology of AD. The dysregulated synaptic elimination is evident in AD pathology, with the previous reports revealing neuronal abnormality, in which there was enhanced C1q expression in synapses[18]. Together, our findings provide evidence that microglial SIRPα signal is inhibited in AD, which significantly accelerates synapses loss by promoting inappropriate synaptic elimination during AD progression. We have demonstrated that such negative regulating signal of synaptic pruning/elimination is dysregulated in neurological diseases, which further contributes to the synaptopathology and neurological symptoms. Since SIRPα-CD47 signal has been widely studied as immune checkpoint pathway, it may also be a potential therapeutic target in synaptopathology related neurological disorders[37,38].

In summary, our findings provide evidence that microglial SIRPα signal negatively regulate synaptic pruning during early neurodevelopment, which is important for maintaining homeostasis in brain. During AD pathology, microglial SIRPα signal is disrupted, which subsequently induces excessive elimination of synapses and enhances cognitive impairment.

## Methods
**Mice and treatment**. All experimental procedures were approved by Model Animal Research Center of Nanjing University and according with Laboratory Animal Care Guidelines. C57BL/6 mice, Cx3cr1[CreERT2] mice, CD47-KO mice, and AD (APPswe and PSEN1dE9 in a single locus) mice were purchased from the Model Animal Research Center of Nanjing University.

SIRPα fl/fl mice were generated by flanking sirpa exon 2–6 with loxP sites. The base pair length between the 2 lox P sites was 14446 bp. Microglial SIRPα-cKO mice were generated after we treated Cx3cr1[CreERT2]: SIRPα[fl/fl] mice with tamoxifen (TAM) at different stages indicated. After TAM induction, PCR genotyping generated a 343-bp fragment suggesting SIRPα gene (exon 2–6) ablation. While before exon 2–6 deletion, the genetic distance between these two primers is ~14 kb, resulting in no bands under normal PCR condition (elongation time is 45 s). All primers used for mice genotyping were listed in Supplementary Table 1. All mice were bred under standard conditions of constant temperature (22 ± 1 °C), humidity (relative, 30%), in a pathogen-free facility and exposed to a 12-h light/dark cycle.

**Tamoxifen injection**. In brief, tamoxifen was dissolved in ethanol then diluted by corn oil. Neonatal mice were injected with 50 μg tamoxifen solution (1 μg/μl) or vehicle into stomach at P1, P2, and P3[39]. Adult mice were injected with tamoxifen intraperitoneally once every 24 h for a total of 5 consecutive days, with the dose of 75 mg tamoxifen/kg body weight (20 μg/μl). Mice were killed for further investigation at several time points as indicated.

**Isoflurane treatment**. P7/P28 mice were exposed to 100% oxygen carrying 1.5% isoflurane for 4 hours. Oxygen was delivered at 5 L/min flow rate controlled by a calibrated flowmeter. In control group, mice were exposed to room air for 4 h. All mice were returned to their original cages together upon regaining righting reflex. One day after isoflurane treatment, mice were euthanatized and brains were taken out for further analysis.

**Stereotaxic injections of Aβ**. For ICV injection, 3 months aged mice (male) were anesthetized and 2 μl Aβ oligomers (1 μg/μl) were injected into the left lateral ventricles. The stereotaxic coordinates were 0.4 mm posterior to the Bregma, 1 mm lateral to the midline, and 2.5 mm ventral to the surface of the dura mater. PBS containing 4% DMSO was injected as control. Mice were killed for further analysis at different time points indicated.

### Cell cultures and treatment
**Primary neuron culture**. Briefly, cerebral cortices were harvested from E14-E16 embryos. The meninges were removed and the cerebral cortices were dissected into small pieces and then digested by 0.025% trypsin for 30 min at 37 °C. Cell suspension were filtered through cell strainer to remove chunks. After centrifugation, cells were plated onto PDL-coated plastic coverslips in 24-well plates for neuron-microglia co-culture, or cells were plated onto PDL-coated plastic coverslips in 6-well plates for protein extraction. The cells were maintained in Neurobasal medium (Invitrogen, Camarillo, CA, USA) with B27 supplement (Invitrogen, Camarillo, CA, USA) and GlutaMax Supplement (Invitrogen, Camarillo, CA, USA) for 14 days before neuron-microglia co-culture. Neurons were cultured for 10–12 days and incubated with TTX (1 μM) for 48 h before immunofluorescent staining or protein extraction. For CD47 expression assay, neurons were stimulated with 0.2 μM Aβ monomer or Aβ oligomer for 24 h before protein analysis.

**Primary microglial culture**. Briefly, cerebral cortices were harvested from neonatal mice before digestion and centrifugation to obtain a pellet for cell seeding. The microglia were harvested from the astrocyte layer 6–10 days later by shaking the flasks at 200 rpm for 1–2 h at 37 °C. The media was centrifuged and the cells were pelleted. Cells were seeded onto poly-D-lysine (PDL)-coated plastic coverslips in 24-well plates (1 × 10⁵ cells/per well) with DMEM/F12 complete medium plus 10% heat-inactivated fetal bovine serum, or the cell suspension were added into neurons directly for subsequent neuron-microglia co-culture.

For lentiviral transduction, EF1a-SIRPα lentivirus expressing system (LV6, 1 × 10⁹ TU/ml, Genepharma, Shanghai, China) was used to overexpress SIRPα in isolated microglia. Cells were exposed to lentivirus (at a multiplicity of infection of five) 3 days after seeding, and incubated for 6 h before the medium was refreshed. Two days later cells were subjected to Aβ stimulation.

Microglia were stimulated with 0.2 μM Aβ monomer or Aβ oligomer for 24 h before protein analysis. For phagocytic assay, cells were subjected to 0.2 μM Aβ42 treatment for 3 h before incubating with synaptosomes.

**Neuron-microglia co-cultures**. Briefly, primary cortical microglia were harvested from the astrocyte layer by shaking the flasks at 200 rpm for 1–2 h at 37 °C, isolated microglia were added into DIV 14 neurons at a 1:3 microglia (6.6 × 10³ cells/per well) to neuron (2 × 10⁴ cells/per well) ratio for 3 days co-culture. Co-cultures were terminated by fixing cells with 4% PFA for further analysis.

**Immunoblotting**. To obtain total proteins, brain tissues or cultured cells were lysed by RIPA lysis buffer (Thermo Scientific, Rockford, IL, USA) supplemented with protease inhibitor, and protein concentration was quantified by BCA protein assay kit (Thermo Scientific, Rockford, IL, USA). The protein samples were loaded onto 10% SDS/PAGE gel before transferred onto poly vinylidene difluoride (PVDF) membranes (Roche Diagnostics, Indianapolis, IN, USA). Then 5% nonfat milk was used to block the membranes before incubation with different primary antibodies overnight at 4 °C. Primary antibodies were used as follows: goat anti-CD47 (1:1000, R&D Systems, AF1866), rabbit anti-SIRPα (1:5000, Abcam, ab8120), rabbit anti SIRPB1 (1:2000, LifeSpan Biosciences, LS-C679465), and mouse anti-GAPDH (1:2000, Santa Cruz Biotechnology, sc-59541). After washing, the membranes were incubated with horseradish peroxidase (HRP)-conjugated secondary antibodies for 1 h. Secondary antibodies were used as follows: donkey anti-rabbit IgG-HRP (1:1000-1:5000, Santa Cruz Biotechnology, sc-2077); donkey anti-mouse IgG-HRP (1:2000-1:5000, Santa Cruz Biotechnology, sc-2314); donkey anti-goat IgG-HRP (1:1000-1:2000, Santa Cruz Biotechnology, sc-2020). Then ECL Western Blotting Detection Kit (Thermo Scientific) was used for color detection. The quantification of protein level was analyzed by ImageJ analysis software. Uncropped gel scans are provided in a Source Data file.

**Immunofluorescence staining**. Mouse brains were collected after transcardial perfusion with phosphate-buffered saline (PBS) and 4% paraformaldehyde (PFA) sequentially. Brains were fixed in 4% PFA for 24 h, then soaked in 15% and 30% sucrose solution for dehydration. Coronal sections of 20 μm or 30 μm thickness were cut in a Leica cryostat (Leica CM1950) for immunofluorescent staining. Afterwards, tissue sections were incubated in 5% bovine serum albumin (BSA) with 0.3% triton-X 100 in PBS for 1 h. The brain sections were incubated with primary antibodies at 4 °C overnight, and then washed three times with PBS before incubated with the appropriate secondary fluorescent antibodies for 1 h at room temperature. Tissue was stained with DAPI and rinsed three times with PBS before image capture using Leica TCS SP8-MaiTai MP confocal microscope. For primary cultured cell immunostaining, cells were fixed with 4% PFA for 10 min, and the following procedure were the same as brain tissues.

Information of antibodies used were as follows: primary antibodies: rabbit anti Iba-1 (1:500, wako, 019-19741); chicken anti-MAP2 (1:1000, Abcam, ab5392); chicken anti-GFAP (1:1000, Abcam, ab4674); mouse anti-SIRPα (1:500, Millipore, MAB1407P); rabbit anti-NeuN (1:1000, Abcam, ab177487); rabbit anti-Homer1 (1:200, Synaptic Systems, 160-003); guinea pig anti-Vglut1 (1:1000, Millipore, AB5905); guinea pig anti-Vglut2 (1:1000, Millipore, AB2251); mouse anti-PSD95 (1:100-1:500, Abcam, ab2723); rabbit anti-Synapsin 1 (1:1000, Millipore, AB1543); mouse anti-Synaptophysin (1:2000, Sigma, S5768); rat anti-CD47(1:200, BD Pharmingen, 555297); and 6E10 (1:500, BioLegend, 803001). Secondary fluorescent antibodies (Thermo Fisher Scientific, 1:1000) were as follows: Goat anti-Rabbit IgG (H + L) Secondary Antibody, Alexa Fluor 488 conjugate (Thermo Fisher Scientific, A-11034); Goat Anti-Rabbit IgG (H + L) Antibody, Alexa Fluor 594 Conjugated (Thermo Fisher Scientific, A-11012); Goat anti-Mouse IgG (H + L) Cross-Adsorbed Secondary Antibody, Alexa Fluor 488 (Thermo Fisher Scientific, A-11001); Goat anti-Chicken IgY (H + L) Secondary Antibody, Alexa Fluor 488 (Thermo Fisher Scientific, A-11039); Goat anti-Rabbit IgG (H + L) Secondary Antibody, Alexa Fluor 633 (Thermo Fisher Scientific, A-21071); Donkey Anti-Rat IgG (H + L) Antibody, Alexa Fluor 594 Conjugated (Thermo Fisher Scientific, A-21209); Goat Anti-Guinea Pig IgG (H + L) Antibody, Alexa Fluor 488 Conjugated (Thermo Fisher Scientific, A-11073); Goat anti-Mouse IgG (H + L) Cross-Adsorbed Secondary Antibody, Alexa Fluor 594 (Thermo Fisher Scientific, A-11005); Goat anti-Chicken IgY (H + L) Secondary Antibody, Alexa Fluor 594 (Thermo Fisher Scientific, A-11042); Donkey anti-Rat IgG (H + L) Cross-Adsorbed Secondary Antibody, Alexa Fluor 488 (Thermo Fisher Scientific, A-21208).

TUNEL staining was performed using TUNEL BrightGreen Apoptosis Detection Kit (A112, Vazyme) according to protocols.

**Eye-specific segregation analysis**. Mice were anesthetized with isoflurane, for injection of neonatal mice, a pair of small scissors was used to open the eyelid to expose the sclera. Afterwards, a sterile 30.5 G needle was used to puncture a small hole in the eye where the sclera begins. Once the vitreous flowed out of the hole, a blunt ended needle attached to a Hamilton syringe was inserted into the hole to inject dye into the eye slowly. Cholera toxin subunit B (CTB) conjugated to Alexa 594 (ThermoFisher Scientific, C-22842) and Alexa 647 (ThermoFisher Scientific, C-34778) were intraocularly injected in the right and left eye (1–2 μl; 0.5% in sterile saline) respectively[40]. Mice were killed 24 h later and brain tissues were fixed in 4% PFA overnight, cryoprotected in 15% and 30% sucrose sequentially, and then sectioned coronally at 40 μm using Leica cryostat (Leica CM1950).

Images were acquired by Leica TCS-SP8 Laser Scanning microscope with ×10 objective, and images were analyzed by ImageJ software using the multi-threshold quantitative method[40]. In each genotype 7 mice were examined and at least four images were analyzed per mouse. The degree of overlap was analyzed by unpaired t test at every threshold level.

**Whole-cell patch clamp recordings on brain slices**. Mice were decapitated after sodium pentobarbital anesthesia, the coronal slices (300 μm thickness) containing hippocampus were obtained by a vibroslicer (VT 1200S, Leica, Wetzlar, Germany) and incubated in artificial cerebrospinal fluid (ACSF, NaCl 125 mM, KCl 3.25 mM, NaH₂PO₄ 1.25 mM, NaHCO₃ 25 mM, MgCl₂·6H₂O 1 mM, D-glucose 11 mM, CaCl₂ 2 mM) with 95% O₂ and 5% CO₂, at 32 ± 0.5 °C for at least 1 h. Then the slices were maintained at room temperature. For whole-cell patch clamp recordings, the slices were transferred to a submerged chamber and continuously perfused with 95% O₂ and 5% CO₂ oxygenated ACSF at a rate of 2 mL/min maintained at 32 ± 0.5 °C.

Whole-cell patch clamp recordings were performed with borosilicate glass pipettes (resistance of 4–6 MΩ) filled with internal solution (140 mM K-methylsulfate, 7 mM KCl, 2 mM MgCl₂, 10 mM HEPES, 0.1 mM EGTA, 4 mM Na₂-ATP, 0.4 mM GTP-Tris, adjusted to pH 7.25 by KOH). The hippocampal CA1 neurons were visualized through Olympus BX51WI microscope (Olympus, Tokyo, Japan) during recording. mEPSCs were recorded at a holding potential of −70 mV with the presence of 1 μM tetrodotoxin (MedChen Express) and 50 mM gabazine (GABAA receptor antagonist; Tocris, Bristol, UK) for at least 1 h through Axopatch-700B amplifier (Axon Instruments, Foster City, CA), and the signals were transferred into a computer through a Digidata-1440A (Axon Instruments, Foster City, CA) for data capture. Clampex 10.7 (Axon Instruments, Foster City, CA) was used for data collection and pClamp10.0 (Axon Instruments, Foster City, CA) was used for data analysis.

**Synaptosome isolation and pHrodo labeling**. Briefly, mice were anesthetized and forebrains were quickly removed and homogenized in ice-cold gradient buffer (320.0 mM sucrose, 5.0 mM HEPES, 0.1 mM EDTA, pH 7.5). The homogenate was centrifuged at $1000 \times g$ for 20 min to collect supernatant, and supernatant was centrifuged at $1200 \times g$ for 10 min to discard cell nuclei and debris and collect supernatant. The supernatant was centrifuged at $10,000 \times g$ for 10 min to obtain pellet. The pellet was resuspended in gradient buffer and loaded onto a sucrose gradient (0.8 M: 1.2 M = 1:1). The thin layer (the layer between 0.8 M sucrose and 1.2 M sucrose) was collected carefully after centrifugation at $100,000 \times g$ for 1 h, and diluted with an equal volume of ultrapure water before centrifuged at $100,000 \times g$ again to acquire the purified synaptosome pellet. The synaptosome pellet was resuspended in ice-cold artificial cerebrospinal fluid (aCSF, 2 mM CaCl₂, 132 mM NaCl, 3 mM KCl, 2 mM MgSO₄, 1.2 mM NaH₂PO₄, 10 mM HEPES, and 10 mM glucose, pH 7.4).

For pHrodo labeling, synaptosomes were incubated with pHrodo Red succinimidyl (NHS) ester (Life Technologies, P36000) or pHrodo Green STP ester (Life Technologies, P36013) in sodium carbonate buffer pH 9.0 for 2 h at 4 °C in the dark at the concentration of 1 μL pHrodo/1 mg synaptosomes. After incubation unconjugated pHrodo was washed by DPBS.

**In vitro microglial engulfment assay**. Equal amount of WT or CD47-KO pHrodo-red conjugated synaptosomes were added to cultured WT or SIRPα KO microglia. After incubation period of 1 h with synaptosomes, cells were washed in warm (37 °C) PBS and fixed in warm 4% PFA for 10 min. After fixation, cells were washed in PBS again, then blocked and permeabilized in 5% bovine serum albumin (BSA) and 0.3% triton-X100 in PBS for 1 h at RT. Cells were incubated with primary antibody overnight at 4 °C and followed by secondary fluorescent antibody staining for 1 h at RT. Cells was staining with DAPI and rinsed three times with PBS before image capture using Leica TCS SP8-MaiTai MP confocal microscope. In all, 5–6 fields of view were collected randomly from each coverslip. When WT or CD47-KO synaptosomes were conjugated to pHrodo-red or pHrodo-green dyes respectively, images should be captured when cells were alive because fluorescence of pHrodo-green is undetectable after fixation. An equal amount of pHrodo-red and green labeled synaptosomes were mixed and added to primary cultured microglia by the amount of 0.017 mg of total synaptosomes. Cells were washed in warm (37 °C) PBS after 1 h of incubation with synaptosomes, then stained live for 15 min at 37 °C with CX3CR1 antibody conjugated to Alexa Fluor 647 (1:100, BioLegend, 149004). Cells were imaged live using Leica TCS SP8-MaiTai MP confocal microscope using LAS X 3.3.0 software.

**CD47 positive synaptic structures labeling in vivo**. For Vglut1/CD47 or PSD95/CD47 double labeling in brain section, z-stack images (at 0.2 μm intervals, 12 images) were imaged using a Zeiss LSM 880 microscope with a ×63 objective and 2× electronic zoom.

For PSD95/CD47/Iba-1 triple labeling in microglia in vivo, z-stack images (at 0.3 μm intervals) were imaged using a Zeiss LSM 880 microscope with a ×63 objective and 2× electronic zoom. In all, 3D-structured illumination images were generated using ZEN 3.1 blue edition software (Zeiss).

**NanoSight detection**. Synaptosomes were incubated with anti-CD47-FITC antibody (1:100, BD Pharmingen, 555298) on ice for 30 min in the dark before detection. Then synaptosomes extracted from mice forebrain were analyzed using NanoSight NS300 (Malvern Instruments) to obtain particle diameter data, particle number and FITC fluorescent information through NanoSight NTA 3.2 software (Malvern Instruments).

**Preparation of single-cell suspension and flow cytometry**. Mice were anesthetic and executed to harvest cerebrums. Cerebrums were dissected into small pieces and digested in papain for 30 min. After digestion, cell suspensions were filtered through 70 μm cell strainer to remove clumps, and cell pellets were resuspended in 30% Percoll solution following centrifugation. The upper myelin layer was discarded, and cell pellets were resuspended in medium (HBSS containing 2% FBS and 1 mM EDTA) to obtain single-cell suspension. Isolated single cells were treated as follows: 15 min incubation with FC-receptor blocker CD16/CD32 antibody (Thermo Scientific, MA5-18012, 1:200) at 4 °C, followed by 30 min incubation with anti-CD45-PE antibody (eBioscience, 12-0451-82, 1:200), anti-SIRPα-FITC antibody (Biolegend, 144006, 1:200), and anti-CD11b-APC antibody (eBioscience, 17-0112-82, 1:200) or rat IgG isotype. For intracellular staining, cells were fixed with 4%PFA for 10 min and permeabilized with PBS containing 0.1% Triton-X 100. Cells were incubated with anti-PSD95 antibody (Cell Signaling Technology, 3450, 1:500) for 30 min, then incubated with secondary fluorescent antibodies for 30 min. FACS analysis was performed on Beckman Coulter Gallios Flow Cytometer with Kaluza 1.0 software (Beckman Coulter). The data were analyzed by Kaluza Analysis 1.5a Software (Beckman Coulter).

**Adult microglia isolation**. Mice were anesthetized and perfused intracardially with ice-cold Dulbecco's phosphate-buffered saline to harvest brains. After the subsequent tissue dissociation, debris removal and red blood cell removal procedures, Adult Brain Dissociation Kit (130-107-677, Miltenyi Biotec) was used to generate single-cell suspension. Microglia were further isolated from the single-cell

suspension using MACS Separation Columns (MS) (130-042-201, Miltenyi Biotec) and magnetic CD11b Microbeads (130-093-634, Miltenyi Biotec).

**Stereotaxic injections of AAV**. Mice were anesthetized and 100 nl AAV (OBIO technology, $1.25 \times 10^{13}$ GC/mL, AAV-hSyn-hM4D(Gi)-mCherry-WPRE) were injected into the primary visual cortex. The stereotaxic coordinates were 3.5 mm posterior to the Bregma, 2.6 mm lateral to the midline, and 0.5 mm ventral to the surface of the dura mater. AAV-hSyn-mCherry-P2A-3xFLAG-WPRE was injected as control. Four weeks later, mice were injected Clozapine-N-oxide (CNO) (Sigma, C0832) intraperitoneally at the dose of 0.5 mg/kg twice a day for consecutive 3 days. Mice were killed for further analysis 4 h after the final injection of CNO. For immunofluorescence, brain tissue was fixed in 4% PFA overnight, cryoprotected in 15% and 30% sucrose before sectioning.

**Aβ preparation**. Briefly, Aβ1-42 peptides were dissolved in hexafluoroisopropanol (HFIP) and was dried in an airing cupboard subsequently. Aβ monomers prepared by dissolving the Aβ in dimethyl sulfoxide (DMSO) at the concentration of 2.2 mM. Aβ oligomers were prepared by dissolving Aβ in DMSO and oligomerized by incubating at room temperature for 48 h. Aβ oligomers were diluted at the concentration of 100 mM (stock solution). Aβ monomers and Aβ oligomers were stored at −80 °C[41].

**Aβ42 ELISA**. Soluble Aβ42 was directly obtained in brain homogenates prepared with ice-cold lysis buffer. To detect insoluble cerebral Aβ42 in brain, insoluble pellets were further extracted in 70% formic acid by sonication and spun at $13,000 \times g$ for 20 min. Samples were neutralized in 1 M Tris buffer. The levels of Aβ42 were quantified using commercially available human ELISA kits (Wako-296-64401) according to manufacturer's guidelines. Data obtained from the cortical homogenates were expressed as nanograms of Aβ content per milligrams of total protein (ng/mg).

**Golgi staining**. Golgi staining was performed using FD Rapid GolgiStain[TM] kit according to its manufacturer's protocol (FD NeuroTechnologies, PK401, Columbia, MD). Briefly, brains were immersed in impregnation solution (a mixture of FD Solution A:B = 1:1) for 2 weeks at room temperature in the dark. Then brains were immersed in FD Solution C and preserved in the dark at room temperature for 48–72 h. The brain tissues were coronally sliced (150 μm) using Leica CM1950 cryostat and mounted on 0.5% gelatin-coated slides. The sections were then stained with the staining solution according to manufacturer's protocol and coverslipped using Permount medium. Images of cortical pyramidal neurons were captured using Olympus BX51 with Cell sens dimension 1.12 software (Olympus) at random.

For pyramidal neuron spine density analysis, the number of spines was counted on the following segments: for apical dendrites, dendritic segments (20 μm) located further than 100 μm from cell soma of pyramidal neurons were quantified.

**Behavioral analysis**

*Open field*. Mice were placed in the center of an empty field (40 × 40 × 24 cm) and allowed to explore freely. Total distance the mice traveled within 5 min was recorded. The maze was cleaned with 70% 2-propanol between the trials.

*Morris water maze*. The procedures consisted of 1 day of the visible platform test and 5 days of the hidden platform test, plus a spatial probe trial. Time taken by the mouse to find and climb onto the platform was recorded as latency. In the probe trial, the percentage of total time in target quadrant was measured. Tracking of animal movement was achieved with a DigBehv-MM tracker system (MobileDatum Co. Ltd, Shanghai, China).

**Fluorescent images quantification**

*Cell number and morphology*. Cell density was quantified by Iba1[+]/NeuN[+]/MAP2[+] cell number in a given area. Cell volume was determined as Iba1[+]/NeuN[+] area per cell. Primary cell purity was determined by MAP2[+] cell/DAPI or Iba1[+] cell/DAPI. Microglial branches in vivo were analyzed as follows. Brain sections stained with

Iba-1 were imaged using Leica TCS SP8-MaiTai MP to obtain z-stack images (at 1 μm intervals). Images were performed z-stack projection and converted into 8-bit. The modified images were binarized and skeletonized before branch analysis using ImageJ-Fiji 2.0.0 software[22]. Circularity of microglia in vitro was obtained by the formula: Circularity = $4\pi$ (area/perimeter$^2$). A circularity value of 1.0 indicates a perfectly circular cell, and values near zero indicate elongated and ramified microglia. All images were processes using ImageJ.

*Synaptic density in vivo*. Briefly, 20 μm sections stained with synaptic markers (Homer1/Vglut1, Homer1/Vglut2, Synapsin1/PSD95) were captured by Leica TCS SP8-MaiTai MP confocal microscope or Zeiss LSM 880 microscope through x63 objective lens. Captured images were used to quantify the number of colocalized pre- and postsynaptic puncta by ImageJ software, and single-channel images were used to quantify single synaptic marker density by ImageJ. Synaptic density was determined as puncta number/given area. Researchers were blinded to genotype during imaging.

*Synaptic density in vitro*. Neuron-microglia co-cultures were quantified by immunofluorescent staining using antibodies for MAP2 (1:1000, Abcam, ab5392), synaptophysin (1:2000, Sigma, S5768), PSD95 (1:250-1:500, Abcam, ab2723), and Iba-1 (1:500, Wako, 019-19741) respectively. Microglia cells were centered for image capture by Leica TCS-SP8 Laser Scanning microscope with ×40 objective. To quantify the density of synaptophysin or PSD95 puncta, concentric circles were drawn around the microglia at increments of 20 μm. The puncta number within each concentric circle were counted by Sholl analysis. Synaptic density was calculated by puncta number/neurites length in the given area. All images were processes using ImageJ. Researchers were blinded to genotype during imaging.

*Plaque and plaque-associated microglia*. Mice brain section was labeled with Iba-1 and 6E10 antibody followed by standard immunostaining procedures[42]. Images were captured using 10× objective and numbers of plaque were counted using ImageJ. For microglia recruitment assay, plaques were selected according to their size and divided into two groups (20–50 μm diameter group and 50–80 μm diameter group). Area of plaque-associated microglia was determined by Iba-1 staining and plaque area was calculated as 6E10[+] area. Ratio of Iba-1[+] area/6E10[+] area in the region of interest were measured by ImageJ.

*Fluorescent intensity*. Images were converted into 8-bit to obtain total gray value in given area by ImageJ software. Fluorescent intensity was calculated as follows: gray value (SIRPα or CD47)/(Iba-1 or NeuN)

*3D Microglial Engulfment quantification*. For microglial engulfment assay, z-stack images (at 0.3 μm intervals) were captured by Leica TCS SP8-MaiTai MP confocal microscope with a 40× or 63× lens. Images were captured by selecting microglia with Iba-1 positive channel randomly without bias. Images were processed by smoothing and subtracting background using ImageJ software. Afterwards, 3D volume surface renderings of each z-stack were created using Imaris 7.4.2 software (Bitplane). The volume of the microglia and synaptic puncta was calculated using surface rendered images. Engulfment percentage was calculated as volume of internalized synaptic puncta/volume of microglial cell. Researchers were blinded to genotype during imaging. To examine CD47[+]/PSD95[+] puncta in microglia, double surface rendering was conducted. The first surface rendering preserved CD47 and PSD95 positive signal inside microglia. The second surface rendering was applied to check colocalization of CD47 and PSD95 signal inside the cell.

*2D microglial engulfment assay in vitro*. Engulfment was analyzed by calculating the fraction of the Iba-1 area that was overlapped by the pHrodo area.

Cells were imaged using Leica TCS SP8-MaiTai MP confocal microscope. Researchers were blinded to genotype during imaging.

**Human tissue samples**. Human frontal lobe tissue lysate (GTX26550 for AD patients; GTX28727 and GTX15360 for controls) was purchased from GeneTex, Inc.

**Table 1 Information of brain tissues from AD patients and control subjects.**

| Cat# | Description | Sample size | Total amount | Concen. (mg/ml) | Age range | Gender |
|---|---|---|---|---|---|---|
| GTX28727 and GTX15360 | Human brain: frontal lobe (normal) tissue lysate | $n = 4$ | 200 μg | 5 | 77–87 | 2 Males |
| | | | 200 μg | 5 | | 2 Females |
| | | | 200 μg | 5 | | |
| | | | 200 μg | 5 | | |
| GTX26550 | Human brain: frontal lobe (Alzheimer's disease) tissue lysate | $n = 5$ | 1 mg | 5 | 73–87 | 3 Males |
| | | | 1 mg | 5 | | 2 Females |
| | | | 1 mg | 5 | | |
| | | | 1 mg | 5 | | |
| | | | 1 mg | 5 | | |

(Irvine, CA). Information about the sample is shown in Table 1 below. The homogenized lysate (20 μg) was mixed with 5× protein loading buffer before boiling at 95 °C for 10 min and was subjected to 10% SDS-PAGE followed by immunoblotting.

**Ethics statement**. The use of human tissue in this study was approved by the Ethics Review Committee of Nanjing Drum Tower Hospital Affiliated to Nanjing University Medical School.

**Statistical analysis**. GraphPad Prism 8 software and GPower 3.1 software were used for statistical analysis. All data were confirmed normal distribution by D'Agostino and Pearson/Shapiro-Wilk/Kolmogorov-Smirnov normality tests. Data were analyzed using one-way ANOVA (followed by Tukey's multiple comparison test or Dunnett's multiple comparisons test), two-way repeated-measures ANOVA (followed by Tukey's multiple comparisons test), two-way ANOVA (followed by Sidak's multiple comparisons test) or unpaired two-tailed $t$ test. When normality of samples failed, Kruskal–Wallis test (followed by Dunn's multiple comparisons) was performed. All $p$ values and statistical test values were indicated in Supplementary Statistical Data. All data with significant difference were analyzed by GPower to obtain effect size and power value (>0.8).

**Reporting summary**. Further information on research design is available in the Nature Research Reporting Summary linked to this article.

## Data availability

All relevant data generated for this study are included in the article/Supplementary Material/Source Data File. Other data/materials (including SIRPα fl/fl mice) that support the findings of this study are readily available from the corresponding author upon reasonable request. (Applicants should sign the material transfer agreement and take charge of all the cost regarding to mouse transfer). Source data are provided with this paper.

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

## Acknowledgements

This work was supported by grants from the Ministry of Science and Technology of China (2018YFA0507100), National Science Foundation of China (31741053 and 81370926), and China Postdoctoral Science Foundation funded project (2020M671449).

## Author contributions

L.L. and K.Z. conceived and designed the experiments. X.D., J.W., M.H. and Z.C. performed the experiments. X.D., J.L. and Q.P.Z. analyzed the data. L.L., C.Y.Z. and Y.X. contributed reagents/materials/analysis tools. K.Z. and L.L. wrote and edited the paper.

## Competing interests

The authors declare no competing interests.
