## [Peer Review File · Nature Communications]

Reviewers' Comments:

Reviewer #1:

Remarks to the Author:

Synaptic pruning is important in normal development but it is becoming increasingly clear that inappropriate synapse elimination due to inflammation or mutation can have dramatic effects on brain homeostasis and behavior. In this manuscript, the authors described the role of the signal regulatory protein SIRP α in regulating microglia engulfment activity during development and neurodegenerative disease progression. SIRP α cooperates with 'do-not-eat-me' signal molecule CD47 to inhibit microglia-dependent synapse elimination. While the SIRP α -CD47 complex has been described in Lehrman et al., 2018, *Neuron* (from Beth Stevens group in 2018), as well as SIRP α knockout mice, the authors used SIRP α microglia specific conditional knockout mouse model to rule out confounding effect due to systematic loss of SIRP α . In addition, the authors extended the knowledge and showed the relevance of the SIRP α -CD47 complex in the progression of Alzheimer's disease both in human samples and in mouse models.

Concerns to be addressed:

- Expression pattern of SIRP α during development
 - o Which cell type(s) express SIRP α ?
 - o How does the expression pattern change over development stages (p5, p30, p60)?
 - o Does it show spatial heterogeneity? (in Lehrman et al. 2018, *Neuron*, fig.1, the authors showed expression level of CD47 peaked and concentrated in the dLGN at p5, which coincided with synaptic pruning in the dLGN. Does SIRP α follow such pattern?)
- Functional relevance of SIRP α /CD47 during development and disease
 - o The authors showed decreased synaptic density in SIRP α cKO dLGN at p30 (supp. Fig3), is the eye-specific retinogeniculate segregation pattern affected in newborn mice as well as in adults (suggested ages: p5, p10 and p30)?
 - o The authors showed aggravated cognitive defects in AD/SIRP α KO mice. While SIRP α KO mice did not show apparent defect in Morris Water Maze escape assay, do they have other behavioral, sensory, or motor defects? In addition, SIRP α is expressed by neurons in adult mice according to Lehrman et al., 2018, is the behavioral defect caused by loss of SIRP α in the neurons and possibly other glial cells? A thorough behavioral study including more tests should be performed on WT, SIRP α KO, microglia cKO, AD, and AD with SIRP α cKO mouse models.
- SIRP β 1 is a closely related member from the same protein family. It shares similar extramembrane domain with SIRP α but has a very different intracellular domain. Instead of interacting with CD47, SIRP β 1 interacts with DAP12. Contrary to SIRP α , SIRP β 1 activates microglia engulfment activity (Zhang et al., 2015, *Brain Research*; Gaikwad S et al., 2009 *Am. J. Pathol.*)
 - o How does SIRP β 1 expression change in the SIRP α cKO microglia?
 - o How does SIRP β 1 expression change in human AD patients, and mouse models for AD?
- In figure 2, the authors showed there was less synaptic structure surrounding primary SIRP α KO microglia than WT microglia. Comments on this experiment:
 - o The percentage of PSD95 or synaptophysin positive microglia in each case should be added.
 - o The amount of synaptic material engulfed by microglia should be quantified.
 - o Local neuronal density should be taken into account during the quantification.
 - o The purity of the primary cell culture should be assessed. At least a nuclei counter stain should be included in the images to show the overall quality of the cell culture.
 - o A significant proportion of PSD95 and Sph puncta were localized relatively far away from Map2 staining. Is this due to image quality? The authors should address this observation to make the data more convincing.
- In figure 5, the authors showed there are both CD47+ and CD47- synaptic structures in primary neurons. In addition, CD47 expression correlated with neuronal activity. The following in vivo experiments should be added:
 - o Repeat the SPH/PSD95/CD47 staining on tissue sections to show presence of both CD47+ and CD47- synaptic structures in vivo.
 - o To directly address whether microglia preferentially engulf 'weak' CD47 negative synapses,

microglia engulfment assay using 3D-reconstruction and surface rendering method should be done on tissue sections by quantifying ratio of CD47+/PSD95+ : CD47-/PSD95+ puncta inside Iba-1 positive microglia comparing adult vs. newborn mice; control vs. mice under anesthesia.

Reviewer #2:

Remarks to the Author:

Ding and Wang et al. investigate the role of microglial SIRPα in regulating microglia-driven synapse elimination during both development and Alzheimer's disease (AD). By using a microglial-specific SIRPα knockout line, the authors demonstrate that the loss of microglial SIRPα signaling through CD47 impairs microglial engulfment of synapses during development. Interestingly, the authors also demonstrate a role for microglial SIRPα in AD synapse loss. While this work fills in a missing piece in the microglial don't-eat-me signaling, there are some concerns with the technical aspects of this work. Additionally, how microglial-specific SIRPα deters microglia-driven synapse loss in AD remains unclear.

1) Mouse line

a. The authors use a tamoxifen inducible cre line to drive the postnatal deletion of SIRPα in microglia. The control for these experiments were mice-treated with just vehicle. While this is a good control, another control that is critical will be SIRPα-f/f mice (no cre) that also receive tamoxifen. Tamoxifen could cause side-effects that may influence synapses.

b. The role of neuronal SIRPα has been established. Therefore, it will be beneficial for authors to demonstrate that levels of SIRPα in other cell types (neurons and astrocytes) are not altered in their mouse line at ages closer to when analyses were done.

2) Elucidating the role of microglial SIRPα in AD is the major strength of this paper. However, the mechanism of microglial SIRPα function in AD is still unclear.

a. Is microglial SIRPα function in AD dependent on CD47? Does Aβ decrease synaptic CD47? Does loss of CD47 also drive AD synapse loss?

b. The AD behavioral and spine loss analysis were done using constitutive SIRPα KO mice. Neuronal SIRPα has been demonstrated to regulate synapses, and SIRPα has a developmental role. Can AD synapse loss be rescued by reintroducing SIRPα specifically into microglia? Does neuronal SIRPα also play a role in AD synapse loss? Authors should use the mice described in Fig 8A (microglia and adult-specific SIRPα KO) in their analysis.

3) What is the rationale for evaluating synapses and engulfment at P5 and P28? Is this when microglial engulfment peaks in the visual cortex/hippocampus? Cortical synapse formation is not complete by P5, therefore the authors cannot conclude that synapse formation is not affected. The absolute number of synapses and more timepoints should be included for both synapse and microglial analyses to conclude that microglial SIRPα regulates synapse elimination. Changes in neuronal cell death, neuronal morphology and axon targeting should also be evaluated before authors can conclude that microglial SIRPα regulates synapses.

4) Based on the data shown in Fig 2, the authors may not conclude that more synaptic elements were "engulfed". At the least, the authors should quantify PSD95 and synaptophysin density in the absence of microglia as well. The authors should also discuss why synapse loss is greater further away from the microglia.

5) How the sample size was determined for each experiment should be specified. Many experiments do not appear to have enough power to conclude. Statistical analysis is not described in the methods. The authors should also standardize and calculate Ns per mouse for immunohistochemistry and engulfment data (in some cases Ns per slices were used).

6) Vglut1 and Homer should be quantified separately as well. From the images, it looks like Vglut1

(presynaptic) changes more than Homer (postsynaptic). Is there a preference for microglia to engulf presynaptic terminals? If so, microglial engulfment assay should be done with Vglut1, not just PSD95. The authors should also use a secondary only control in the engulfment assays to demonstrate that the engulfed PSD95+ puncta are indeed synapses.

7) In Fig 7, authors should include data for controls (WT mouse and SIRPa KO in non-AD background).

8) The authors need to quantify to show that microglial morphology (volume and Scholl analysis for branching) does not change. This is important especially because in Fig 2, microglia morphology looks different.

9) Isoflurane's mechanism of action is unclear. Other mechanisms of manipulating activity (such as DREADDs) may also be used.

10) "Cognitive impairment" may not be concluded from just escape latency times. The authors should include data to rule out motor or visual deficits.

Minor:

- 1) The authors should correct the grammar used in this paper.
- 2) Higher magnification images should be shown especially for images showing changes to dendritic spines.

Reviewer #3:

Remarks to the Author:

Summary:

The article under review investigates the role of CD47-SIRPa signaling pathway in synapse pruning. The key role of microglia in synapse remodeling in both neurodevelopment and adulthood is now well established. Previous data showed the potential involvement of SIRPa in synaptic pruning, as it is a receptor of CD47 that produces "do not eat me" signals (Lehrman et al., 2018) as the authors noted. This submitted article employs a wide range of different techniques and experimental approximations both in vitro and in vivo. The authors show a relation between microglial ablation of SIRPa and a reduction in synaptic density during development and in mouse models of AD. They present very nice experiments demonstrating a decline in SIRPa in AD mouse models as behavior performance declines, that KO of SIRPa increases pruning and accelerated behavior deficits. There are several nice complementary experiments such as showing loss of microglial SIRPa increases A β -induced synapse loss and promoting microglia mediated synaptic elimination. In vitro SIRPa deficient microglia ingested more synaptosomes than wild type microglia. Furthermore, they show studies manipulating CD47 that support their premise that SIRPa-CD47 signaling regulates the clearance of inactive or injured synapses (suppression of neural activity reduces CD47 expression). They also present some data about the role of SIRPa during Alzheimer's disease progression although addition of more human samples would strengthen this association (see below). Their findings suggest a potential role of microglial SIRPa in synaptic pruning under physiological and pathological conditions. However, the connection between SIRPa decline and cognitive decline was overstated in even mouse models due to the lack of synaptic data in the aged (8 mo) WT and SIRPa knock outs, as mentioned below. The comments below are all fixable. This is a timely and important paper, but some of the details of the analysis and gaps in direct mechanistic pathways need to be clarified to be of value to the field.

Major issues:

1. In the discussion, the authors refer to "C1q-CR3" signaling. That is incorrect and highly misleading for the field as C1q does not bind CR3. All such text must be changed to either iC3b/C3b-CR3 signaling or C1q-dependent iC3b/C3b-CR3 signaling.

2. While Supplemental Figure 1f demonstrates the dependence of SIRPa deletion on Tam treatment, the authors should include in the figure or text the base pair number between the 2 lox P sites since that makes a difference in the validity of the Cx3CR1-Creert2 mice control mice (-TAM) per Van Hove, et al (Van Hove et al., 2020). In addition, the location of the PCR primer pairs should be noted on the diagram in Sup Fig 1a, to enable evaluation of Sup Fig 1e. Ie. Why no band in the TAM- with those primer pairs? Finally, the TdTomato experiment does not prove that the SIRP1a floxed gene is deleted (Van Hove et al., 2020), and thus those panels can be deleted from the figure.

3. For some immunofluorescence staining of synapses, the criteria for measuring a synapse must be written in the methods. For example, in Sup Fig 3?? [Also, there is no scale bar for the P5 panels in Sup Fig.3 – must be added. In addition, for some synapses, the authors employed VGlut1 antibody, yet, for other staining they used VGlut2. Since these are often seen on different cells, please explain the reason why they changed from VGlut1 to VGlut2?

4. Regarding the increment in synaptic engulfment by SIRPa deficient microglia, authors do not mention how they chose the cells to assess with a confocal microscope. Are the researchers blinded to the genotype of samples? Do they use an unbiased method for recording microglial cells? This question is very important and authors should explain it in the method section of the paper.

5. As for the regulation of synaptic CD47 by neural activity, authors use TTX and isoflurane to prove that less active synapses express lower levels of CD47. However, the connection between isoflurane, TTX and less active synapses should be explained or discussed in more detail, so the reader could actually understand why the authors are testing this hypothesis.

6. In the section discussing microglial SIRPa downregulation in AD pathology with human samples, a larger number of samples should be used as the sex/age/previous pathologies could make a big difference in these tissues. While N=3 is usually acceptable for mice, because AD pathology is much more stable, in humans a higher number of patients and control samples should be presented.

7. While Fig. 6 panels e-l provide compelling data that oligomers inhibit SIRPa, panels 6c and 6d require some explanation as the behavior of the 6 M animals does not align as described with the rest of the figure, as a big increase of SIRPa protein in WT mice at 6 months. Do the authors have an explanation for this upregulation? Were all the blots and samples done on the same day? It appears that this was done only once. Was the big increase in SIRPa at 6 mo confirmed in wild type? Have they tried to repeat this experiment with more WT mice? This huge difference in WT animals at 6 months does not make sense, especially when the flow cytometry assay showed the same levels of SIRPa microglia between WT mice at 2, 5 and 8 months old. In relation to cytometry analysis, why did the authors change from 6 and 12 months old (protein level assays) to 5 and 8 months old (flow cytometry assays)? This suggests different cohorts? Some discussion of this large difference (panels c and d vs the rest of the panels) must be included (and validated). Regarding IHC of SIRPa and Iba1, the paper only shows the results in 5 months old mice. If the objective of this experiment is to confirm the decrease of SIRPa levels in the brain, more/all different ages of mice should be assessed (and shown).

8. Additionally, a control group without any injections should be included for the stereotaxic amyloid injection experiments. ICV injections could start an inflammatory response in the brain and the effect of this response should be evaluated when comparing control mice with PBS injection and control mice without any injections. Also, information about what mice (age and genotype) were used for this ICV injections experiments is missing. Also, how long after the injections were the mice analyzed? What was the concentration and volume of mAb and oAb injections? For the in vitro studies (Fig 6i-k) how much time between the addition of the amyloid and analysis of the SIRPa?

9. SIRPa deficiency enhanced cognitive impairment and synaptic loss without modifying A β plaque deposition: it would be useful to perform ELISAs/MSD to test if there is any soluble/insoluble A β in the brain of these mice (preceding plaque deposition) that can correlate with the cognitive deficit they showed.

10. Synaptic density does not seem to decrease between 5 and 8 months of age, at which time the APP mice are losing cognitive performance. While lack of decrease in the synaptic density may be

due to different scorers, at least the WT and WT SIRPa at those ages must be added to provide context. Does synaptic density correlate with loss of cognition (Fig. 7 b and c versus Fig. 7m, 7n and sup Fig 4 a and b)? (Perhaps not, and this would be an impactful finding if true.) This need for WT data (and KO only) is also important for the % synapsis throughout the paper. (Fig. 7 k and l, sup Fig 4 c and d.), especially since at 8 mo the SIRPa are losing behavior performance to the same extent as the AD mice in Fig. 7c). In addition, in this same section, information about the quantification of synapses in the cortex and hippocampus is missing. Did they use ImageJ, Imaris or something else?

11. Regarding Iba1/6E10 staining, authors count manually the number of microglia around amyloid plaques. Because not all the plaques are of the same size and because microglia forms clusters around these plaques, it is very difficult to distinguished between different microglia. Therefore, this quantification method may not be very reliable/meaningful. An additional analysis of the area covered by microglia in the region of interest correlated to the area covered by 6E10 in the same area would resolve this issue.

12. Supplemental Figure 2 is the same as Sup. Fig. 5a-e. Sup. Figure 5f-g could be incorporated into Sup Fig. 2 and noted in the text with the rest of Sup. Fig 2. Again, information about neuronal and microglial density quantification method is missing. I suggest the authors include a section in Methods with all the information related to quantitative analysis. Neuronal and microglial analysis were in the cortex. As the paper has previously shown some results in the hippocampus, authors should perform this analysis in both regions.

13. Referring to Figure 8h-k, additional information about this in vitro experiment is needed. Since this is in vitro, it does not relate to Fig. 8a, and thus perhaps moving this to supplemental material would aid the reader. (The Fig 8 in vivo demonstrates the point, and thus the in vitro is good to show but is not critical to the main points of the manuscript.)

14. Finally, authors wrote: "In vitro assay revealed that A β oligomers stimulation increased phagocytosis of synaptosomes in primary microglia" but the graph shows that not only A β increases the phagocytosis but also the monomers in a lower level. This must be discussed and a rationale proposed to explain this.

Minor:

1. Although the article is in general well-written, there are some grammatical errors in the text ("The initial number of synapses in P5 mice did not alter significantly...", "After A β injection, the number of synapses significantly reduced..." or "Several studies have demonstrated that complementary signal C1q-CR3..." are only a few examples). All must be revised. All "evidences" should be changed to "evidence".

2. Typo? In the concentration of Ab oligomers – written in the methods as at 100 mM. Even 100uM is high. Must be revised and also importantly the final concentration and final amount of both mAb and oAb for each experiment must be provided.

3. Non-AD individuals are referred to as normal people, which leads to think people with Alzheimer's disease are not normal. Use an appropriate word to referred to these controls (non-demented, non-AD, controls, Braak ...).

4. In the abstract, "resulting" should be replaced with "correlating": "...microglial SIRPa expression declines in the progression of Alzheimer's disease (AD), resulting in excessive microglia mediated synapse elimination as well as aggravated cognitive dysfunction."

Lehrman, E. K., Wilton, D. K., Litvina, E. Y., Welsh, C. A., Chang, S. T., Frouin, A., . . . Stevens, B. (2018). CD47 Protects Synapses from Excess Microglia-Mediated Pruning during Development. *Neuron*, 100(1), 120-134 e126. doi:10.1016/j.neuron.2018.09.017

Van Hove, H., Antunes, A. R. P., De Vlaminck, K., Scheyltjens, I., Van Ginderachter, J. A., & Movahedi, K. (2020). Identifying the variables that drive tamoxifen-independent CreERT2 recombination: Implications for microglial fate mapping and gene deletions. *Eur J Immunol*, 50(3), 459-463. doi:10.1002/eji.201948162

Point-to-point response

Reviewer #1:

Synaptic pruning is important in normal development but it is becoming increasingly clear that inappropriate synapse elimination due to inflammation or mutation can have dramatic effects on brain homeostasis and behavior. In this manuscript, the authors described the role of the signal regulatory protein SIRP α in regulating microglia engulfment activity during development and neurodegenerative disease progression. SIRP α cooperates with 'do-not-eat-me' signal molecule CD47 to inhibit microglia-dependent synapse elimination. While the SIRP α -CD47 complex has been described in Lehrman et al., 2018, *Neuron* (from Beth Stevens group in 2018), as well as SIRP α knockout mice, the authors used SIRP α microglia specific conditional knockout mouse model to rule out confounding effect due to systematic loss of SIRP α . In addition, the authors extended the knowledge and showed the relevance of the SIRP α -CD47 complex in the progression of Alzheimer's disease both in human samples and in mouse models.

Concerns to be addressed:

- Expression pattern of SIRP α during development

o Which cell type(s) express SIRP α ?

Response: Our data show that SIRP α is expressed in neuron and microglia but not in astrocyte (new sfig1 a), which is consistent with the previous report (*new ref 30*).

o How does the expression pattern change over development stages (p5, p30, p60)?

Response: Immunostaining and flow cytometry analysis demonstrated higher level of microglial SIRP α at P5, while it started to decline at P30 and maintained low level at P60 (new sfig1 c, e-g). In the meantime, neuronal SIRP α expression displayed a time-dependent increase during development stage (new sfig1 b, d), which is consistent with the previous report that SIRP α promotes synapse maturation (*new ref 30*).

o Does it show spatial heterogeneity? (in Lehrman et al. 2018, *Neuron*, fig.1, the authors showed expression level of CD47 peaked and concentrated in the dLGN at p5, which coincided with synaptic pruning in the dLGN. Does SIRP α follow such pattern?)

Response: Although SIRP α levels changed during development, we did not find obvious spatial heterogeneity of SIRP α expression in P5 or P30 mouse brains (new sfig1 h). The increased fluorescence intensity in P30 mouse brains may be the result of enhanced neuronal SIRP α expression (new sfig1 h).

- Functional relevance of SIRP α /CD47 during development and disease

o The authors showed decreased synaptic density in SIRP α cKO dLGN at p30 (supp. Fig3), is the eye-specific retinogeniculate segregation pattern affected in newborn mice as well as in adults (suggested ages: p5, p10 and p30)?

Response: In our developmental study, microglial SIRP α was specifically deleted in neonatal mice (*Cx3cr1^{CreERT2}; SIRP α ^{fl/fl}*) by early tamoxifen induction (P1-P3). We analyzed eye-specific retinogeniculate segregation pattern, and found that it was not affected at P5 but the overlap between ipsilateral and contralateral signals at P10 and P30 was significantly decreased, respectively (new sfig7 e-h).

o The authors showed aggravated cognitive defects in AD/SIRP α KO mice. While SIRP α KO mice did not show apparent defect in Morris Water Maze escape assay, do they have other behavioral, sensory, or motor defects? In addition, SIRP α is expressed by neurons in adult mice according to Lehrman et al., 2018, is the

behavioral defect caused by loss of SIRP α in the neurons and possibly other glial cells? A thorough behavioral study including more tests should be performed on WT, SIRP α KO, microglia cKO, AD, and AD with SIRP α cKO mouse models.

Response: We appreciate reviewer's constructive comments/suggestions. We did not detect any behavioral abnormality of SIRP α KO mice under normal condition. However, these mice are hypersensitive to inflammatory stress, as stereotactic injection of LPS (low dose) into lateral ventricular cause sustained overactivation of microglia, neuronal loss as well as decreased locomotor activity (unpublished data). We have been working on the long-term AD model using AD/SIRP α cKO mice (microglial SIRP α was specifically ablated at 2 months' age by tamoxifen induction). In the revised manuscript, we have used this mouse model to exclude the potential impact of neuronal SIRP α deficiency as well as the developmental change caused by early microglial SIRP α deletion (new Fig. 7). In our revision, all results derived from SIRP α conventional KO mice are removed from the revision. In the new study about AD/SIRP α cKO mice, we have included open field test and modified our Water Maze testing procedures with 1 day of the visible platform test and 5 days of the hidden platform test, plus a spatial probe trial. Open field analysis showed similar locomotor activity among all those mouse groups (Control; SIRP α cKO; AD; AD/SIRP α cKO) at 5-months age (new Fig. 7b). Although we observed decreased locomotor activity in mice with AD background at 8-months age, there were no significant differences between AD and AD/SIRP α cKO mice (new Fig. 7b). In addition, all mice showed similar escape latencies in the visible platform test (new Fig. 7c), implying comparable vision among these mice. Hidden platform test and spatial probe trial results suggested that microglial SIRP α deficiency in adulthood accelerated memory declination in AD mice (new Fig. 7, d and e).

- SIRP β 1 is a closely related member from the same protein family. It shares similar extramembrane domain with SIRP α but has a very different intracellular domain. Instead of interacting with CD47, SIRP β 1 interacts with DAP12. Contrary to SIRP α , SIRP β 1 activates microglia engulfment activity (Zhang et al., 2015, Brain Research; Gaikwad S et al., 2009 Am. J. Pathol)

- o How does SIRP β 1 expression change in the SIRP α cKO microglia?

Response: SIRP β 1 protein level was not altered in SIRP α cKO mice brain (new sFig. 5, a and b). QPCR analysis further demonstrated that mRNA level of SIRP β 1 remained unchanged in microglia after SIRP α deletion (new sFig. 5c).

- o How does SIRP β 1 expression change in human AD patients, and mouse models for AD?

Response: Western-blot analysis showed that SIRP β 1 expression was elevated both in AD patients and AD mice brain. QPCR analysis showed mRNA level of SIRP β 1 increased in microglia of AD mice (new sFig. 5d-h). Although SIRP β 1 regulates microglia phagocytic activity and plays an important role in AD (new refs 32 and 33), it differs from SIRP α which interacts with CD47 and controls/inhibits the selective engulfment of microglia via “do not eat me” signal.

- In figure 2, the authors showed there was less synaptic structure surrounding primary SIRP α KO microglia than WT microglia. Comments on this experiment:

- o The percentage of PSD95 or synaptophysin positive microglia in each case should be added.

- o The amount of synaptic material engulfed by microglia should be quantified.

- o Local neuronal density should be taken into account during the quantification.

- o The purity of the primary cell culture should be assessed. At least a nuclei counter stain should be included in the images to show the overall quality of the cell culture.

Response: According to Reviewer's instructions, we have included these quantifications in our revision.

1. We found that most of microglia were PSD95/synaptophysin positive (>90%) after co-culturing (new sFig. 9g), which showed no significant differences in both groups.
2. The amount of synaptic material engulfed by microglia was further quantified by 3D reconstruction and surface rendering after immunostaining (Iba-1 with PSD95/SPH), which revealed that SIRP α deficient microglia engulfed more synaptic structures (new Fig. 2e-g).
3. The overall neuronal density was identical between two groups of cells. In the revised manuscript, neurites length in the given area was taken into account during quantification. We recorded synaptic density as puncta number/neurites length in the given circle (new Fig. 2b-d).
4. Iba-1/MAP2 staining revealed that purify of primary microglia/neurons are high (>90%) before co-culturing (new sFig. 9, a and b).

o A significant proportion of PSD95 and Sph puncta were localized relatively far away from Map2 staining. Is this due to image quality? The authors should address this observation to make the data more convincing.

Response: We greatly thank reviewer for pointing out this. MAP2 signal was relatively low in *original Figure 2*, which did not label all the neurites especially those small ones. In the revise manuscript, we have reconducted this experiment and improved MAP2 staining condition to label the neurites. As shown in new Fig. 2b, most of PSD95 and SPH puncta were localized along with the labeled neurites.

• In figure 5, the authors showed there are both CD47⁺ and CD47⁻ synaptic structures in primary neurons. In addition, CD47 expression correlated with neuronal activity. The following *in vivo* experiments should be added:

o Repeat the SPH/PSD95/CD47 staining on tissue sections to show presence of both CD47⁺ and CD47⁻ synaptic structures *in vivo*.

Response: As our SPH antibody is not applicable for *in vivo* staining, we have included images of Vglut1/CD47 or PSD95/CD47 double labeling on brain section, which showed both CD47⁺ and CD47⁻ synaptic structures *in vivo* (new Fig. 5c).

o To directly address whether microglia preferentially engulf 'weak' CD47 negative synapses, microglia engulfment assay using 3D-reconstruction and surface rendering method should be done on tissue sections by quantifying ratio of CD47⁺/PSD95⁺: CD47⁻/PSD95⁺ puncta inside Iba-1 positive microglia comparing adult vs. newborn mice; control vs. mice under anesthesia.

Response: We performed the additional experiments accordingly. As the peak of microglia mediated synaptic pruning occurs at the early stage of development (to eliminate redundant synapses) and it becomes relatively rare in adulthood under normal condition, we assessed synaptic engulfment by microglia in cortex at P28. Using 3D-reconstruction and surface rendering methods, we have quantified CD47⁺/PSD95⁺ puncta in microglia from SIRP α -cKO and control mice. In control microglia, most of PSD95⁺ puncta were CD47 negative, indicating the preferential engulfment of CD47⁻ synapses. Meanwhile, the amount of CD47⁺ synaptic structure was significantly increased in SIRP α deficient microglia, suggesting that the selective engulfment of CD47⁻ synapses is compromised in microglia after losing SIRP α (new Fig. 5, n and o).

In mice under anesthesia condition, microglia engulfed more PSD95⁺ synaptic structures (new sFig. 12c and d). However, volume ratio of CD47⁺/PSD95⁺ puncta to total synaptic structures decreased compared to the mice under normal condition (new sFig.12c and e). This may be the result of decreased synaptic CD47 expression after neural activity inhibition, which leading to more CD47 negative synaptic structures to be

eliminated by microglia.

Reviewer #2:

Ding and Wang et al. investigate the role of microglial SIRP α in regulating microglia-driven synapse elimination during both development and Alzheimer's disease (AD). By using a microglial-specific SIRP α knockout line, they demonstrate that the loss of microglial SIRP α signaling through CD47 impairs microglial engulfment of synapses during development. Interestingly, the authors also demonstrate a role for microglial SIRP α in AD synapse loss. While this work fills in a missing piece in the microglial don't-eat-me signaling, there are some concerns with the technical aspects of this work. Additionally, how microglial-specific SIRP α deters microglia-driven synapse loss in AD remains unclear.

1) Mouse line

a. The authors use a tamoxifen inducible cre line to drive the postnatal deletion of SIRP α in microglia. The control for these experiments were mice-treated with just vehicle. While this is a good control, another control that is critical will be SIRP α -f/f mice (no cre) that also receive tamoxifen. Tamoxifen could cause side-effects that may influence synapses.

Response: We appreciate reviewer's constructive suggestion. In the revised version, we have included SIRP $\alpha^{fl/fl}$ mice receiving tamoxifen as another control. The results showed that our conclusions regarding to synaptic morphology were not changed when we compared SIRP α cKO mice with this non-Cre control (new Fig. 1). Notably, in mice with Cx3cr1^{creERT2} background, insertion of the Cre and EYFP knocks out one copy of endogenous CX3CR1. It is reported that Cx3cr1^{+/-} may affect microglial cytokine production as well as neuronal function in adult mice (*new ref 14*). Therefore, in the following study in AD, we used Cx3cr1^{CreERT2}:SIRP $\alpha^{fl/fl}$ mice treated with vehicle as normal control.

b. The role of neuronal SIRP α has been established. Therefore, it will be beneficial for authors to demonstrate that levels of SIRP α in other cell types (neurons and astrocytes) are not altered in their mouse line at ages closer to when analyses were done.

Response: We have examined SIRP α expression in several cell types and found it mainly expressed in neuron and microglia (new sFig. 1a). Neuronal level of SIRP α are not altered in SIRP α cKO mice (P28) compared to control (new sFig. 3, e and f).

2) Elucidating the role of microglial SIRP α in AD is the major strength of this paper. However, the mechanism of microglial SIRP α function in AD is still unclear.

a. Is microglial SIRP α function in AD dependent on CD47? Does A β decrease synaptic CD47? Does loss of CD47 also drive AD synapse loss?

Response: We thank reviewer for pointing out this important question. In the revised manuscript, we have demonstrated that synaptosome CD47 level declined in the late stage of AD (8 Months) (new sFig.16, c and d). Besides, CD47 KO mice were more vulnerable upon A β stimulation, which showed greater loss of synapses and increased microglial engulfment compared to WT control (new sFig.16 h-k). These results are consistent to that in SIRP α cKO mice (new Fig. 8), indicating that SIRP α and CD47 may work together in the synaptic pathology of AD.

b. The AD behavioral and spine loss analysis were done using constitutive SIRP α KO mice. Neuronal SIRP α

has been demonstrated to regulate synapses, and SIRP α has a developmental role. Can AD synapse loss be rescued by reintroducing SIRP α specifically into microglia? Does neuronal SIRP α also play a role in AD synapse loss? Authors should use the mice described in Fig 8A (microglia and adult-specific SIRP α KO) in their analysis.

Response: Indeed, using constitutive SIRP α KO mice can't rule out the potential effects of neuronal SIRP α deficiency in AD synaptic loss. In order to exclude the potential impact of neuronal SIRP α as well as the developmental change caused by early microglial SIRP α deletion, we have used AD^{APP^{swe},PSEN1^{dE9}}/SIRP α cKO mice model (by tamoxifen induction, microglial SIRP α was specifically ablated at 2 months' age) to investigate the role of microglial SIRP α deficiency on AD pathogenesis in adulthood (new Fig. 7a). In addition, we have included open field test and modified our Water Maze testing procedures with 1 day of the visible platform test and 5 days of the hidden platform test, plus a spatial probe trial. Open field analysis showed similar locomotor activity among all those mice groups (Control; SIRP α cKO; AD; AD/SIRP α cKO) at 5-months age (new Fig. 7b). Although we observed decreased locomotor activity in mice with AD background at 8-months age, there was no significant difference between AD and AD/SIRP α cKO mice (new Fig. 7b). In addition, all groups of mice showed similar escape latencies in the visible platform test (new Fig. 7c), implying comparable vision among these mice. Hidden platform test and spatial probe trial results suggested that microglial SIRP α deficiency in adulthood accelerated memory declination in AD mice (new Fig. 7, d and e). Besides, we have demonstrated that synaptic loss in AD/SIRP α cKO mice was significantly increased (Fig. 7l-o). As rescue experiment can underline the importance of microglial function in AD pathology, we have successfully overexpressed SIRP α in microglia *in vitro*. As shown in new sFig. 15, c and d, overexpression of SIRP α in microglia reduced microglial engulfment towards synaptosomes after A β treatment. Unfortunately, when we tried to overexpress SIRP α in microglia *in vivo* using lentiviruses or adeno-associated viruses (Cre-dependent) system, we found the expression efficiency was extremely low (~10%). Therefore, we did not continue the rescue experiment *in vivo*.

3) What is the rationale for evaluating synapses and engulfment at P5 and P28? Is this when microglial engulfment peaks in the visual cortex/hippocampus? Cortical synapse formation is not complete by P5, therefore the authors cannot conclude that synapse formation is not affected. The absolute number of synapses and more timepoints should be included for both synapse and microglial analyses to conclude that microglial SIRP α regulates synapse elimination.

Response: Microglial engulfment of synaptic structures in V1 cortex was analyzed at P28, which is around the peak of the critical period for experience-dependent plasticity. Theoretically, reduced synapses we observed in SIRP α cKO mice at P30 could be an accumulative result caused either by neuronal growth defect or by enhanced microglial pruning. We have demonstrated that synaptic density is not altered at P5 or P15 in V1 cortex of SIRP α cKO mice, while it significantly decreases at P30 compared to control (new Fig. 1b, d-f). The initial synaptic number at P5 or P15 did not alter significantly, indicating that synaptic reduction in SIRP α cKO mice at P30 is less likely due to neuron growth defect. Notably, synaptic density in hippocampus of SIRP α cKO mice showed remarkable decrease at P15 and P30, suggesting that the over-pruning period in hippocampus of SIRP α cKO mice may occur earlier than that in visual cortex (new Fig. 1c, g-i). In the revised version, we have included P15 as an addition time point and used absolute number to quantify synaptic density (new Fig. 1b-i and Fig. 4b-i).

Changes in neuronal cell death, neuronal morphology and axon targeting should also be evaluated before authors can conclude that microglial SIRP α regulates synapses.

Response: We agree with reviewer on this. In the revision, we have compared neuronal status of SIRP α cKO mice with control groups under normal condition. We found no significant changes of neuronal density or volume in cortex and hippocampus in SIRP α cKO mice (new sFig. 3a-d). TUNEL staining also showed little apoptotic signal in these mice (new sFig. 3g). As we mentioned above, synaptic density in visual cortex did not change in SIRP α cKO mice at P5 or P15 while it significantly decreased at P30. Besides, SIRP α cKO and control mice displayed similar innervation of the dLGN in P5 (new sFig. 7i-j). These data suggest that reduced synapses are less likely to be the result of neuron growth defect.

4) Based on the data shown in Fig 2, the authors may not conclude that more synaptic elements were “engulfed”. At the least, the authors should quantify PSD95 and synaptophysin density in the absence of microglia as well. The authors should also discuss why synapse loss is greater further away from the microglia.

Response: Accordingly, we have reconducted this experiment and quantified PSD95/ synaptophysin density in areas that are absent of microglia, which showed that the overall synaptic density was not altered after adding a few microglia with different genotype in the primary neuron culture (new sFig. 9, c and d). Additionally, we quantified fluorescent signal of synaptic markers (PSD95/SPH) inside microglia using 3D-reconstruction and surface rendering method, which revealed that there were more synaptic structures in SIRP α deficient cells (new Fig. 2e-g). Regarding to reviewer’s second concern, we re-checked our original data and quantification methods. In the original manuscript, data was misrepresented as absolute number of synaptic puncta (which should be **number of synaptic puncta/neurites length in the given circle**) in several concentric circles with different radius. Please see the Table 1 below, which listed absolute numbers of synaptic puncta of PSD95 in different concentric circles (original data). The reduced number became larger as the radius of circle increased. The misrepresented data (**absolute number instead of density**) may lead to the misinterpretation that synapse loss is greater further away from the microglia. In the revision, we have fixed this problem by quantifying synaptic density (number of synaptic puncta/neurites length in the given circle) in different concentric circles (40, 60, 80 μ m), which shows greater synaptic loss in SIRP α deficient group compared to control (new Fig. 2b-d).

Table 1. Number of PSD95 puncta in different concentric circles

Radius of concentric circle (μ m)	Control	SIRP α KO	Reduced number
30	4.9	1.2	3.7
40	12	4.5	7.5
50	22.7	12.1	10.6
60	38.1	23.1	15
70	58.2	37.2	21

5) How the sample size was determined for each experiment should be specified. Many experiments do not appear to have enough power to conclude. Statistical analysis is not described in the methods. The authors should also standardize and calculate Ns per mouse for immunohistochemistry and engulfment data (in some cases Ns per slices were used).

Response: Technically, sample size is estimated by the significance level (0.05), effect size (base on the data in our previous study) and a given power (usually 0.8). We can also calculate the power value using G-power software. In the original version, there were several experiments with power less than 0.8: Fig. 1e (0.77), Fig. 1g (0.78), Fig. 1i (0.76), Fig. 4g (0.75), Fig. 5L (0.78) and Fig. 7l (0.79). In the revised manuscript, we have

included new control group (new Fig. 1, SIRP α ^{f/f} mice with TAM), expanded sample size (new Fig. 1k and sFig. 12b) or re-calculate data using Ns per mouse (new Fig. 1d-i and Fig. 4d-i), which generated greater effect size and power value (>0.8). Regarding to those new included data, we have conducted power analysis after the experiment and found enough power value (>0.8) in each statistical analysis. Also, we have described our statistical methods with more detail in the Methods section. All statistical data are included in excel as supplementary file (new Supplemental Table 1). Also, we standardize and calculate Ns per mouse in our immunohistochemistry and engulfment assay.

6) Vglut1 and Homer should be quantified separately as well. From the images, it looks like Vglut1 (presynaptic) changes more than Homer (postsynaptic). Is there a preference for microglia to engulf presynaptic terminals? If so, microglial engulfment assay should be done with Vglut1, not just PSD95. The authors should also use a secondary only control in the engulfment assays to demonstrate that the engulfed PSD95+ puncta are indeed synapses.

Response: According to reviewer's comment, we have quantified Vglut1 and Homer1 respectively and found that both pre- and post-synaptic marker decreased significantly in SIRP α cKO mice as well as in CD47 KO mice (new sFig. 6a-f and sFig. 10). Statistical analysis did not show any preference for microglia engulfment towards pre- or post-synaptic structure, either in SIRP α cKO mice or in CD47 KO mice (new sFig. 6a-f and sFig. 10). In addition, our 3D reconstruction data with Vglut1 or PSD95 labeling demonstrated similar results of microglial engulfment assay *in vivo* that SIRP α deficient microglia engulfed more synaptic structures (new Fig. 1, o and p; sFig. 6, g and h). In order to avoid potential misunderstanding cause by Vglut1 staining (P30) in *original figure 1b*, we have replaced it with more typical image (new Fig. 1b). We also included a negative control labeled without primary antibody (only with secondary antibody) to show the specificity of synaptic marker labeling (new sFig. 8).

7) In Fig 7, authors should include data for controls (WT mouse and SIRP α KO in non-AD background).

Response: As mentioned in Q2, we have used AD/SIRP α cKO mice model to evaluate the role of microglial SIRP α in AD pathogenesis. Four groups of mice (Control; SIRP α cKO; AD; AD/ SIRP α cKO) were included in the assessment of behavior (new Fig. 7b-e) as well as synaptic morphology (new Fig. 7l-o). Since those control mice have non-AD background, they were not shown in the plaque or A β analysis (new Fig. 7f-k).

8) The authors need to quantify to show that microglial morphology (volume and Scholl analysis for branching) does not change. This is important especially because in Fig 2, microglia morphology looks different.

Response: According to reviewer's instruction, we have assessed microglial density and branches *in vivo* showing no significant differences between SIRP α cKO and control mice (new sFig. 4). When we reconducted *in vitro* experiment related to Fig. 2, we assessed microglial morphology (volume and circularity) in co-culture and found no remarkable changes between different groups of cells (new sFig. 9, e and f).

9) Isoflurane's mechanism of action is unclear. Other mechanisms of manipulating activity (such as DREADDs) may also be used.

Response: We thank reviewer for pointing out this. Accordingly, we have applied DREADDs in the experiment. In brief, we overexpressed hM4D and mCherry in cortical neuron in adult mice using AAV transduction system (AAV-hSyn-hM4D(Gi)-mCherry) (new sFig. 11). After inhibiting the neuron activity by CNO administration, we measured the protein level of CD47 in those neurons and found remarkable decrease

compared to control (new Fig. 5, i and j).

10) “Cognitive impairment” may not be concluded from just escape latency times. The authors should include data to rule out motor or visual deficits.

Response: Indeed, escape latency alone is not sufficient to support our conclusion. As mentioned in Q2, we have included open field test and modified our Water Maze testing procedures with 1 day of the visible platform test and 5 days of the hidden platform test, plus a spatial probe trial. Open field and visible platform test demonstrated similar locomotor activity and vision between AD and AD/SIRP α cKO mice (new Fig. 7, b and c). Hidden platform test and spatial probe trial results suggested that microglial SIRP α deficiency in adulthood accelerated memory declination in AD mice (new Fig. 7, d and e).

Minor:

1) The authors should correct the grammar used in this paper.

Response: We have requested the help from Professional Language Service to improve the writing of our revision.

2) Higher magnification images should be shown especially for images showing changes to dendritic spines.

Response: We have improved our experimental condition and provided new images with better quality in our revised manuscript.

Reviewer #3:

Summary:

The article under review investigates the role of CD47-SIRP α signaling pathway in synapse pruning. The key role of microglia in synapse remodeling in both neurodevelopment and adulthood is now well established. Previous data showed the potential involvement of SIRP α in synaptic pruning, as it is a receptor of CD47 that produces “do not eat me” signals (Lehrman et al., 2018) as the authors noted. This submitted article employs a wide range of different techniques and experimental approximations both in vitro and in vivo. The authors show a relation between microglial ablation of SIRP α and a reduction in synaptic density during development and in mouse models of AD. They present very nice experiments demonstrating a decline in SIRP α in AD mouse models as behavior performance declines, that KO of SIRP α increases pruning and accelerated behavior deficits. There are several nice complementary experiments such as showing loss of microglial SIRP α increases A β -induced synapse loss and promoting microglia mediated synaptic elimination. In vitro SIRP α deficit microglia ingested more synaptosomes than wild type microglia. Furthermore, they show studies manipulating CD47 that support their premise that SIRP α -CD47 signaling regulates the clearance of inactive or injured synapses (suppression of neural activity reduces CD47 expression). They also present some data about the role of SIRP α during Alzheimer’s disease progression although addition of more human samples would strengthen this association (see below). Their findings suggest a potential role of microglial SIRP α in synaptic pruning under physiological and pathological conditions. However, the connection between SIRP α decline and cognitive decline was overstated in even mouse models due to the lack of synaptic data in the aged (8 mo) WT and SIRP α knock outs, as mentioned below. The comments below are all fixable. This is a timely and important paper, but some of the details of the analysis and gaps in direct mechanistic pathways need to be clarified to

be of value to the field.

Major issues:

1. In the discussion, the authors refer to “C1q-CR3” signaling. That is incorrect and highly misleading for the field as C1q does not bind CR3. All such text must be changed to either iC3b/C3b-CR3 signaling or C1q-dependent iC3b/C3b-CR3 signaling.

Response: Thanks for the reminding. We have made amendments in our revised manuscript according to your suggestion.

2. While Supplemental Figure 1f demonstrates the dependence of SIRP α deletion on Tam treatment, the authors should include in the figure or text the base pair number between the 2 lox P sites since that makes a difference in the validity of the Cx3CR1-CreERT2 mice control mice (-TAM) per Van Hove, et al (Van Hove et al., 2020). In addition, the location of the PCR primer pairs should be noted on the diagram in Sup Fig 1a, to enable evaluation of Sup Fig 1e. Ie. Why no band in the TAM- with those primer pairs? Finally, the TdTomato experiment does not prove that the SIRP1a floxed gene is deleted (Van Hove et al., 2020), and thus those panels can be deleted from the figure.

Response: Indeed, Van Hove *et al.* (new ref. 27) have done very detailed analysis of tamoxifen-independent CreERT2 recombination, which may help us to avoid “leakage” problem at initial designing stage. Based on their results, genetic distance between *loxP* sites is critical. There are less than 3% leaky expression when the length of the floxed sequence is larger than 2.5kb. In our SIRP α cKO model, the genetic distance between 2 loxP sites is 14.4kb (new sFig. 2a), suggesting little impact of “leakage” problem. PCR and Flow cytometry analysis showed no significant TAM-independent SIRP α deletion in microglia (new sFig. 2, c and d). Accordingly, PCR primers for genotyping is noted in sFig. 2a. The length of positive band for SIRP α deficient cell sample is 343bp. In TAM- mice, the genetic distance between PCR primers are around 14kb, producing no bands under normal PCR condition (elongation time is 45 second). TdTomato experiment is conducted at the age of P5, suggesting that early tamoxifen treatment (P1-P3) is efficient for inducing DNA recombination. As it does not prove that the SIRP α floxed gene is deleted, we have removed it from the revised version according to reviewer’s suggestion.

3. For some immunofluorescence staining of synapses, the criteria for measuring a synapse must be written in the methods. For example, in Sup Fig 3?? [Also, there is no scale bar for the P5 panels in Sup Fig.3 – must be added. In addition, for some synapses, the authors employed VGlut1 antibody, yet, for other staining they used VGlut2. Since these are often seen on different cells, please explain the reason why they changed from VGlut1 to VGlut2?

Response: We apologize for the missing information. In the revised manuscript, we have included details for measuring a synapse in method section. The scale bar of P5 mice in *original sFig. 3* is the same as that in P15 and P30 (new sFig. 7a). As VGlut1 is the major isoform in the cerebral cortex and hippocampus, we used VGlut1/Homer1 double labeling to assess synaptic density in these regions at different time points (P5, P15 and P30) (new Fig. 1 and Fig. 4). However, we found little VGlut1 signal in dLGN at P5 while VGlut2 signal is apparent in this area at all timepoints we observed. Therefore, we assessed synaptic density in dLGN using Vglut2 (new sFig7 a). Notably, Lehrman *et al.* (new ref. 9) have demonstrated a decrease in retinogeniculate synapses (Vglut2 positive) in dLGN in CD47 KO mice while corticogeniculate synapses (Vglut1 positive) are not altered. This may be due to the late time (P12) at which corticogeniculate input innervates the LGN (SIRP α -CD47 signal declined in dLGN at this stage).

4. Regarding the increment in synaptic engulfment by SIRP α deficient microglia, authors do not mention how they chose the cells to assess with a confocal microscope. Are the researchers blinded to the genotype of samples? Do they used an unbiased method for recording microglial cells? This question is very important and authors should explain it in the method section of the paper.

Response: In the present study, key results were repeated by at least two co-authors independently. In synaptic engulfment assay as well as synaptic density analysis (both *in vitro* and *in vivo*), researchers were blinded to genotype during imaging. Therefore, cells or scope fields were randomly chosen with no bias for further analysis. This was mentioned in the Methods section of our revision.

5. As for the regulation of synaptic CD47 by neural activity, authors use TTX and isoflurane to prove that less active synapses express lower levels of CD47. However, the connection between isoflurane, TTX and less active synapses should be explained or discuss in more detail, so the reader could actually understand why the authors are testing this hypothesis.

Response: We appreciate reviewer's constructive suggestion. In the revised manuscript, we have used three methods (TTX, DREADD, and isoflurane) to mimic the condition of less active neurons. We examined CD47 level and found it significantly decreased after neuron activity was blocked (new Fig. 5f-l). We have explained these connections with more detail in the result section (Red in "Synaptic CD47 is regulated by neural activity").

6. In the section discussing microglial SIRP α downregulation in AD pathology with human samples, a larger number of samples should be used as the sex/age/previous pathologies could make a big difference in these tissues. While N=3 is usually acceptable for mice, because AD pathology is much more stable, in humans a higher number of patients and control samples should be presented.

Response: We totally agree with reviewer that the result will be more convincing with more human samples. In the present study, we have collected two more AD human samples and one more non-AD sample. New data is presented in new Fig. 6, a and b, of the revision.

7. While Fig. 6 panels e-l provide compelling data that oligomers inhibit SIRP α , panels 6c and 6d require some explanation as the behavior of the 6 M animals does not align as described with the rest of the figure, as a big increase of SIRP α protein in WT mice at 6 months. Do the authors have an explanation for this upregulation? Were all the blots and samples done on the same day? It appears that this was done only once. Was the big increase in SIRP α at 6 mo confirmed in wild type? Have they tried to repeat this experiment with more WT mice? This huge difference in WT animals at 6 months does not make sense, especially when the flow cytometry assay showed the same levels of SIRP α microglia between WT mice at 2, 5 and 8 months old. In relation to cytometry analysis, why did the authors change from 6 and 12 months old (protein level assays) to 5 and 8 months old (flow cytometry assays)? This suggests different cohorts? Some discussion of this large difference (panels c and d vs the rest of the panels) must be included (and validated).

Response: The *original Fig. 6 (c and d)* was the western-blot analysis of SIRP α level in brain tissue, which included both microglial and neuronal SIRP α expression. In our recent study (new ref. 22), we have demonstrated similar results that SIRP α level in brain tissue increased in WT mice at 6-month age (compared to 1-month age). Neuronal SIRP α may contribute to such increase since microglial SIRP α did not change during this period (new Fig. 6, c and d). As other assays in Fig. 6 all detect or deal with microglia-specific SIRP α expression (instead of both microglial and neuronal), we have moved *original Fig. 6 (c and d)* to the supplementary data section (as new sFig. 13, a and b) in the revision to avoid the potential misleading. In this

study, we have demonstrated remarkable declination of SIRP α level in brain tissue of AD mice compared to same aged control (6Mo and 12Mo, sFig. 13, a and b). In order to assess the specific change of microglial SIRP α in AD, we conducted cytometry analysis and immunostaining which revealed specific decrease of microglial SIRP α in AD mice (5Mo and 8Mo age, new Fig. 6c-f and sFig. 13c-f). Most of our studies in AD section (behavioral and morphological analysis) have applied mice at 5-months or 8-months' age, which represents before and after plaque deposition, respectively (new Fig. 7).

Regarding IHC of SIRP α and Iba1, the paper only shows the results in 5 months old mice. If the objective of this experiment is to confirm the decrease of SIRP α levels in the brain, more/all different ages of mice should be assessed (and shown).

Response: The immunostaining of SIRP α and Iba1 at different timepoints (2, 5, 8 months) were included in the revision, which showed that microglial SIRP α declined in 5M and 8M old AD mice compared to same aged control mice (new Fig. 6, e and f; sFig.13c-f).

8. Additionally, a control group without any injections should be included for the stereotaxic amyloid injection experiments. ICV injections could start an inflammatory response in the brain and the effect of this response should be evaluated when comparing control mice with PBS injection and control mice without any injections. Also, information about what mice (age and genotype) were used for this ICV injections experiments is missing. Also, how long after the injections were the mice analyzed? What was the concentration and volume of mAb and oAb injections? For the in vitro studies (Fig. 6, i-k) how much time between the addition of the amyloid and analysis of the SIRP α ?

Response: According to your advice, in the ICV injection experiment, we have included a new group with no injections as control. It showed no significant differences when compared to PBS injection group (new Fig. 6 i, j). We have also provided detailed information (mice age and genotype, drug concentration, experimental duration) of these experiments in related figure legend and Methods section.

9. SIRP α deficiency enhanced cognitive impairment and synaptic loss without modifying A β plaque deposition: it would be useful to perform ELISAs/MSD to test if there is any soluble/insoluble A β in the brain of these mice (preceding plaque deposition) that can correlate with the cognitive deficit they showed.

Response: In order to elucidate the specific role of microglial SIRP α in AD pathogenesis, we have used AD/SIRP α cKO mice model (by tamoxifen induction, microglial SIRP α was specifically ablated at 2 months' age) to exclude the potential impact of neuronal SIRP α deficiency as well as the developmental change caused by early microglial SIRP α deletion. In the revision, we have examined soluble/insoluble A β at different time points (before/after plaque deposition) by ELISA, which reveals no significant differences between AD and AD/SIRP α cKO mice at same age (new Fig. 7, h and i). Based on these data, it can be postulated that microglial SIRP α deficiency enhances cognitive impairment and synaptic loss without modifying amyloid pathology.

10. Synaptic density does not seem to decrease between 5 and 8 months of age, at which time the APP mice are losing cognitive performance. While lack of decrease in the synaptic density may be due to different scorers, at least the WT and WT SIRP α at those ages must be added to provide context. Does synaptic density correlate with loss of cognition (Fig. 7 b and c versus Fig. 7m, 7n and sup Fig 4 a and b)? (Perhaps not, and this would be an impactful finding if true.) This need for WT data (and KO only) is also important for the % synapsis throughout the paper. (Fig. 7k and l, sFig. 4, c and d.), especially since at 8 mo the SIRP α are losing

behavior performance to the same extent as the AD mice in Fig. 7c). In addition, in this same section, information about the quantification of synapses in the cortex and hippocampus is missing. Did they use ImageJ, Imaris or something else?

Response: We appreciate reviewer's constructive comments/suggestions. In the revision, we have included data of control and SIRP α cKO mice with non-AD background (new Fig. 7l-o). By comparing with non-AD background control, we found that synaptic density in AD mice was not significantly altered at the age of 5M, while they displayed remarkable decline of synaptic number at the age of 8M (new Fig. 7l-o). Together, these results suggest that cognitive loss in AD mice (8M) correlates with decreased synaptic density. Images were analyzed by ImageJ. Detailed information about quantification of synapses were included in the Methods section of the revised manuscript.

11. Regarding Iba1/6E10 staining, authors count manually the number of microglia around amyloid plaques. Because not all the plaques are of the same size and because microglia forms clusters around these plaques, it is very difficult to distinguish between different microglia. Therefore, this quantification method may not be very reliable/meaningful. An additional analysis of the area covered by microglia in the region of interest correlated to the area covered by 6E10 in the same area would resolve this issue.

Response: We are grateful for reviewer's suggestion. Accordingly, we have modified our methods for microglia recruitment assay. Plaques were selected according to their size and divided into two groups (20~50 μ m diameter group and 50~80 μ m diameter group). Area of plaque associated microglia was determined by Iba-1 staining and plaque area was calculated as 6E10⁺ area. Ratio of Iba-1⁺ area/6E10⁺ area in the region of interest were measured by ImageJ. The data showed that microglia recruitment around plaques was comparable between AD and AD/SIRP α cKO mice (new Fig. 7 j, k).

12. Supplemental Figure 2 is the same as Sup. Fig. 5a-e. Sup. Figure 5f-g could be incorporated into Sup Fig. 2 and noted in the text with the rest of Sup. Fig. 2. Again, information about neuronal and microglial density quantification method is missing. I suggest the authors include a section in Methods with all the information related to quantitative analysis. Neuronal and microglial analysis were in the cortex. As the paper has previously shown some results in the hippocampus, authors should perform this analysis in both regions.

Response: *Original sFig2* was neuronal and microglial morphology assay in **SIRP α cKO mice**. *Original sFig5* showed neuronal and microglial morphology in **conventional SIRP α KO mice**, which has been removed from the revised edition. Accordingly, we analyzed neuronal and microglial morphology both in cortex and hippocampus (new sFig. 3a-d and sFig. 4). Detailed information about quantification were included in the Methods section (*Fluorescent images quantification*).

13. Referring to Figure 8h-k, additional information about this in vitro experiment is needed. Since this is in vitro, it does not relate to Fig. 8a, and thus perhaps moving this to supplemental material would aid the reader. (The Fig 8 in vivo demonstrates the point, and thus the in vitro is good to show but is not critical to the main points of the manuscript.)

Response: We have moved these panels into supplementary data (new sFig. 15) according to reviewer's suggestion.

14. Finally, authors wrote: "In vitro assay revealed that A β oligomers stimulation increased phagocytosis of synaptosomes in primary microglia" but the graph shows that not only A β o increases the phagocytosis but also the monomers in a lower level. This must be discussed and a rationale proposed to explain this.

Response: We have performed additional tests which confirmed that A β monomers treatment increased microglial phagocytosis at a lower but significant level (new sFig.15, a and b). As we have demonstrated that A β monomers did not alter microglial SIRP α level (new Fig. 6, g and h), such effect may be the result of microglia activation. In the meantime, A β oligomers treatment decreased microglial SIRP α expression (new Fig. 6, g and h), which was associated with significantly enhanced microglial phagocytosis compared to A β monomers treated group. This result suggests that downregulation of SIRP α after A β o treatment further increases microglial phagocytosis of synaptosomes. We have included this explanation in the results section (Red in “Loss of microglial SIRP α increased A β o-induced synapse loss by promoting microglia mediated synaptic elimination”).

Minor:

1. Although the article is in general well-written, there are some grammatical errors in the text (“The initial number of synapses in P5 mice did not alter significantly...”, “After A β injection, the number of synapses significantly reduced...” or “Several studies have demonstrated that complementary signal C1q-CR3...” are only a few examples). All must be revised. All “evidences” should be changed to “evidence”.

Response: We have made proper amendment according to reviewer’s comments.

2. Typo? In the concentration of Ab oligomers – written in the methods as at 100 mM. Even 100uM is high. Must be revised and also importantly the final concentration and final amount of both mA β and oA β for each experiment must be provided.

Response: “100 mM” is the concentration of stock solution of A β oligomers. We have made this clear in the revision. The exact concentration of A β used *in vitro/in vivo* are provided in the methods section and related figure legends.

3. Non-AD individuals are referred to as normal people, which leads to think people with Alzheimer’s disease are not normal. Use an appropriate word to referred to these controls (non-demented, non-AD, controls, Braak ...).

Response: We have used “non-AD vs. AD” in the revised manuscript accordingly.

4. In the abstract, “resulting” should be replaced with “correlating”: “...microglial SIRP α expression declines in the progression of Alzheimer’s disease (AD), resulting in excessive microglia mediated synapse elimination as well as aggravated cognitive dysfunction.”

Response: We agree that the word “correlating” is more suitable and accurate, and have made the replacement accordingly.

Reviewers' Comments:

Reviewer #1:

Remarks to the Author:

Overall the authors very carefully addressed the initial concerns

Additional comment: In order to look for spatial heterogeneity of Sirpa expression, the authors chose to show a 10X mosaic image of full coronal section. This type of imaging does not necessarily provide enough information as the overall fluorescent intensity at this resolution could be affected by background. Suggestion: take high magnification/resolution images at several broad brain regions, especially cortex, dLGN and hippocampus. And the expression level should be reported both as % of Sirpa positive cells and fluorescent intensity.

michael carroll

Reviewer #2:

Remarks to the Author:

The revised manuscript has reasonably addressed most of the concerns to strengthen their findings on the role of microglial SIRPa. However, the following points still remain to be addressed.

1. The absolute number of synapses is still increasing in WT mice from P15 to P30, both in V1 and CA1 (Figures 1 and 4). Therefore, synapses are still being formed between P15 and P30. Even in SIRPacKO and CD47KO mice, the number of synapses increases from P15 to P30. Therefore, the authors cannot conclude: "reduced synapses are more likely due to microglial over-pruning rather than neuron growth defect" or "¹¹SIRPa-CD47 signal axis negatively regulates synapses elimination in brain during early neurodevelopment". SIRPa-CD47 may also affect synapse formation in addition to synapse elimination. Indeed, microglia has been implicated in synapse formation. The authors should clearly state this possibility and revise the conclusions.

2. NeuN staining is not appropriate to assess neuronal morphology. It only indicates that the number of neurons and the size of the cell body are not changed. The authors should revise the conclusion. To assess neuronal morphology, labeling of axons/dendrites/spines with fluorescent proteins (or similar strategies) would be needed.

3. Discussion:

i) Please add the explanation (discussion) as to why SIRPacKO did not show behavioral changes relative to Control.

ii) In SIRPacKO, A β cannot decrease SIRPa anymore. Please discuss the possible effects of A β in SIRPacKO.

4. Methods: How was colocalization defined?

5. Figures: Figure 5H needs representative images.

6. References: Bialas et al., has been retracted. Please remove this citation.

7. There are still typos that should be corrected.

Reviewer #3:

Remarks to the Author:

The authors have responded well to most of the concerns and comments I previously raised.

However, in response to my concern that "In the abstract, 'resulting' should be replaced with 'correlating': "...microglial SIRP α expression declines in the progression of Alzheimer's disease (AD), resulting in excessive microglia mediated synapse elimination as well as aggravated cognitive dysfunction."

The authors Response was: We agree that the word "correlating" is more suitable and accurate, and have made the replacement accordingly. However, that was not changed in the abstract. They have data to support that in AD models but not in AD itself. The change should be made.

The minor comment I have is concerning the inclusion of non-AD animals in all measures such as the synaptic density in Figure 7 (vs. AD only mice +/- SIRP α deletion). While in figure 7, they now have included data of non-AD and non-AD SIRP α cKO mice with non-AD background (new Fig. 7I-o)(and show correlation with the behavior at 5 and 8 mo), this was only done in the cortex, and the data on the AD animal hippocampus has been removed. Since the hippocampus more closely aligns with cognitive loss, I am surprised that that region was not selected for the quantification.

However, the authors provide substantial evidence and supporting experiments to support the major conclusion of the manuscript that microglial SIRP α interacts with CD47 at the synapse and that the loss of that signaling induces/allows synaptic pruning/clearance in development and AD models, and this loss of synaptic density is correlated with loss of behavioral function. A large number of sophisticated and careful measurements were performed in various models to provide conclusive evidence. This is of significance for the field and will have an impact on subsequent investigations in this critical area, with potential to translate into therapeutic target identification.
Andrea J. Tenner

Reviewer #1 (Remarks to the Author):

Overall the authors very carefully addressed the initial concerns

Additional comment: In order to look for spatial heterogeneity of Sirpa expression, the authors chose to show a 10X mosaic image of full coronal section. This type of imaging does not necessarily provide enough information as the overall fluorescent intensity at this resolution could be affected by background. Suggestion: take high magnification/resolution images at several broad brain regions, especially cortex, dLGN and hippocampus. And the expression level should be reported both as % of Sirpa positive cells and fluorescent intensity.

Michael Carroll

Answer: Thank you for the suggestion. To assess whether there is spatial heterogeneity of microglial SIRP α , we conducted Iba-1/SIRP α double labeling and take high magnification images in cortex, dLGN and hippocampus (new sFig 1 h). Expression level of microglial SIRP α is reported as fluorescent intensity of SIRP α /Iba-1 in ROI (Iba-1 positive area). Quantitative analyses showed there was no obvious spatial heterogeneity of microglial SIRP α expression in P5 or P30 mice brain (sFig 1 i and j). (Highlighted in red in page 5, 1st paragraph)

Reviewer #2 (Remarks to the Author):

The revised manuscript has reasonably addressed most of the concerns to strengthen their findings on the role of microglial SIRP α . However, the following points still remain to be addressed.

Q1. The absolute number of synapses is still increasing in WT mice from P15 to P30, both in V1 and CA1 (Figures 1 and 4). Therefore, synapses are still being formed between P15 and P30. Even in SIRP α cKO and CD47KO mice, the number of synapses increases from P15 to P30. Therefore, the authors cannot conclude: "reduced synapses are more likely due to microglial over-pruning rather than neuron growth defect" or "SIRP α -CD47 signal axis negatively regulates synapses elimination in brain during early neurodevelopment". SIRP α -CD47 may also affect synapse formation in addition to synapse elimination. Indeed, microglia has been implicated in synapse formation. The authors should clearly state this possibility and revise the conclusions.

Answer: We have demonstrated that microglia engulfed more synaptic structure in SIRP α cKO and CD47-KO mice during development, which suggests SIRP α -CD47 signal axis protects synapses from excessive elimination mediated by microglia (highlighted in red in page 7, 3rd paragraph).

Regarding to its impact on synapses formation, we have demonstrated that synaptic density (V1 cortex or hippocampus) is not altered in SIRP α cKO or CD47-KO mice at P5, which suggests synapses formation are not affected by SIRP α -CD47 signal deficiency at this time point (Fig. 1b-i and Fig. 4b-i). However, as you suggested, based on the results that number of synapses increases from P15 to P30, we cannot rule out the possibility that deficiency of SIRP α -CD47 signal affects synaptic formation, which may also contribute to the synaptic reduction at this duration. We have made proper amendments of our conclusion (highlighted in red in page 6, 2nd paragraph) and clearly stated of this possibility in our revised edition (highlighted in red in page 12, 1st paragraph in discussion).

Q2. NeuN staining is not appropriate to assess neuronal morphology. It only indicates that the number of neurons and the size of the cell body are not changed. The authors should revise the conclusion. To assess neuronal morphology, labeling of axons/dendrites/spines with fluorescent proteins (or similar strategies) would be needed.

Answer: We agree with your opinion. NeuN staining shows the number of neurons and the size of the cell body are not changed. We have revised our conclusion accordingly (highlighted in red in page 5, 3rd paragraph).

Q3. Discussion:

i) Please add the explanation (discussion) as to why SIRP α cKO did not show behavioral changes relative to Control.

Answer: In the behavioral study in AD section, microglial SIRP α was ablated at

2-months old by tamoxifen administration to exclude the potential impact of early SIRP α deficiency. The pruning peak during early development is over at this stage. As a result, microglial SIRP α deficiency can no longer affect early synaptic pruning and shows little impact on synaptic density at later stages (Fig. 7l-o). Consistently, these SIRP α -cKO mice did not show any behavioral changes as well. We have included this content in the discussion of revised manuscript (highlighted in red in page 13, 2nd paragraph).

ii) In SIRP α cKO, A β cannot decrease SIRP α anymore. Please discuss the possible effects of A β in SIRP α cKO.

Answer: It is reported that A β induces aberrant increase and synaptic localization of C1q, promoting microglia mediated synapses loss in AD. In the present study, we have demonstrated that A β repressed microglial SIRP α expression. As we have provided evidence that SIRP α deficiency increases microglia mediated synaptic pruning during early development, we believe downregulation of microglial SIRP α may further induce excessive synaptic elimination in AD. To verify our hypothesis, we compared synaptic pathology between AD mice and AD+ SIRP α -cKO mice, which showed SIRP α deficiency increased microglia mediated synaptic loss in AD.

In AD+ SIRP α -cKO mice, A β acted as an initial insult that triggers neurotoxic cascade. In addition, it also induces microglia mediated synaptic loss by activating C1q-dependent iC3b/C3b-CR3 signaling pathway. Our results suggest that lack of microglial SIRP α signal may further exacerbate the situation. Notably, ablation of microglial SIRP α in adult mice (2-months old) shows little impact on synaptic density in normal condition. Meanwhile, it induces excessive synaptic elimination during AD pathology. These data suggest SIRP α signal modulates the synaptic removal in a passive manner. SIRP α deficiency just releases the brake of microglia, it takes effect only when the synaptic elimination process is initiated (physiologically or pathologically). We have included this content in the discussion of revised manuscript (highlighted in red in page 15, 1st paragraph).

Q4. Methods: How was colocalization defined?

Answer: For 2D image (Fig. 1b and c for example), we used Image J to set threshold of intensity and pixels to extract positive signals (actual area > 0.15 μm^2) in a given channel. After merging images from different channels, we further get the colocalized signals. If the colocalized area is larger than 0.09 μm^2 , it is recorded as a positive colocalization. For 3D reconstruction image (new Fig. 5o for example), we use Imaris to analyze colocalization by surface rendering and further extract positive signal by set threshold of colocalized voxels (actual volume > 0.03 μm^3).

Q5. Figures: Figure 5H needs representative images.

Answer: Accordingly, we have included representative images related to Figure 5H (new Fig 5h, i).

Q6. References: Bialas et al., has been retracted. Please remove this citation.

Answer: Thank you for reminding us. We have removed this citation and modified related content in introduction and discussion in the new edition (page 3, 2nd paragraph and page 14, 1st paragraph).

Q7. There are still typos that should be corrected.

Answer: Thanks for pointing it out. We have carefully examined the manuscript and made several corrections.

Reviewer #3 (Remarks to the Author):

The authors have responded well to most of the concerns and comments I previously raised.

Q1: However, in response to my concern that "In the abstract, 'resulting' should be replaced with "correlating": "...microglial SIRP α expression declines in the progression of Alzheimer's disease (AD), resulting in excessive microglia mediated synapse elimination as well as aggravated cognitive dysfunction." The authors Response was: We agree that the word "correlating" is more suitable and accurate, and have made the replacement accordingly. However, that was not changed in the abstract. They have data to support that in AD models but not in AD itself. The change should be made.

Answer: We are truly sorry for such carelessness and have replaced the word "resulting" with "correlating" in the abstract of revised manuscript.

Q2: The minor comment I have is concerning the inclusion of non-AD animals in all measures such as the synaptic density in Figure 7 (vs. AD only mice +/- SIRP α deletion). While in figure 7, they now have included data of non-AD and non-AD SIRP α cKO mice with non-AD background (new Fig. 7l-o)(and show correlation with the behavior at 5 and 8 mo), this was only done in the cortex, and the data on the AD animal hippocampus has been removed. Since the hippocampus more closely aligns with cognitive loss, I am surprised that that region was not selected for the quantification.

Answer: Regarding to the analysis of AD/SIRP α -cKO mice, we have provided data of PSD95/SynapsinI double labeling and Golgi staining in cortex in Figure7 (l-o). In addition, we also quantified synaptic density (PSD95/SynapsinI staining) in hippocampus (sfig. 14), which showed similar results suggesting that microglial SIRP α deficiency enhanced synaptic loss in AD mice.

However, the authors provide substantial evidence and supporting experiments to support the major conclusion of the manuscript that microglial SIRP α interacts with CD47 at the synapse and that the loss of that signaling induces/allows synaptic pruning/clearance in development and AD models, and this loss of synaptic density is correlated with loss of behavioral function. A large number of sophisticated and careful measurements were performed in various models to provide conclusive evidence. This is of significance for the field and will have an impact on subsequent investigations in this critical area, with potential to translate into therapeutic target identification.

Andrea J. Tenner

Reviewers' Comments:

Reviewer #2:

Remarks to the Author:

We have no further comments on the manuscript. Our concerns have been addressed in the revised manuscript.

Hisashi Umemori, Nicole Scott-Hewitt (Stevens lab), Sivapratha Nagappan-Chettiar (Umemori lab)

Reviewer #3:

Remarks to the Author:

The authors have responded well to all previous concerns and comments and have provided clear data on some parameters influencing microglia engulfment of synapses in both postnatal development, an AD model and other perturbation of the system. The observations of the role of CD47 and SIRPα on microglial engulfment of synaptic material is of significance for the field and will have an impact on subsequent investigations in this critical area.